# Implications of improved representations of plant respiration in a changing climate

Chris Huntingford[1], Owen K. Atkin[2,3], Alberto Martinez-de la Torre[1], Lina M. Mercado[1,4], Mary A. Heskel[5], Anna B. Harper[6], Keith J. Bloomfield[2], Odhran S. O'Sullivan[2], Peter B. Reich[7,8], Kirk R. Wythers[7], Ethan E. Butler[7], Ming Chen[7], Kevin L. Griffin[9], Patrick Meir[2,10], Mark G. Tjoelker[8], Matthew H. Turnbull[11], Stephen Sitch[4], Andy Wiltshire[12] & Yadvinder Malhi[13]

Land-atmosphere exchanges influence atmospheric $CO_2$. Emphasis has been on describing photosynthetic $CO_2$ uptake, but less on respiration losses. New global datasets describe upper canopy dark respiration ($R_d$) and temperature dependencies. This allows characterisation of baseline $R_d$, instantaneous temperature responses and longer-term thermal acclimation effects. Here we show the global implications of these parameterisations with a global gridded land model. This model aggregates $R_d$ to whole-plant respiration $R_p$, driven with meteorological forcings spanning uncertainty across climate change models. For pre-industrial estimates, new baseline $R_d$ increases $R_p$ and especially in the tropics. Compared to new baseline, revised instantaneous response decreases $R_p$ for mid-latitudes, while acclimation lowers this for the tropics with increases elsewhere. Under global warming, new $R_d$ estimates amplify modelled respiration increases, although partially lowered by acclimation. Future measurements will refine how $R_d$ aggregates to whole-plant respiration. Our analysis suggests $R_p$ could be around 30% higher than existing estimates.

[1] Centre for Ecology and Hydrology, Wallingford, Oxfordshire OX10 8BB, UK. [2] Division of Plant Sciences, Research School of Biology, The Australian National University, Building 134 Canberra, ACT 2601, Australia. [3] ARC Centre of Excellence in Plant Energy Biology, Research School of Biology, The Australian National University, Building 134 Canberra, ACT 2601, Australia. [4] College of Life and Environmental Sciences, Amory Building, University of Exeter, Rennes Drive Exeter, EX4 4RJ, UK. [5] The Ecosystems Center, Marine Biological Laboratory, 7 MBL Street Woods Hole, MA 02543, USA. [6] College of Engineering, Mathematics and Physical Sciences, Laver Building, University of Exeter, North Park Road Exeter, EX4 4QF, UK. [7] Department of Forest Resources, University of Minnesota, 1530 Cleveland Avenue North St Paul, MN 55108, USA. [8] Hawkesbury Institute for the Environment, Western Sydney University, Penrith, NSW 2751, Australia. [9] Department of Earth and Environmental Sciences, Lamont-Doherty Earth Observatory, Columbia University, Palisades, NY 10964-8000, USA. [10] School of Geosciences, University of Edinburgh, Edinburgh, EH9 3FF, UK. [11] Centre for Integrative Ecology, School of Biological Sciences, University of Canterbury, Private Bag 4800, Christchurch, New Zealand. [12] Met Office, FitzRoy Road Exeter, Devon EX1 3PB, UK. [13] School of Geography and the Environment, Oxford University Centre for the Environment, University of Oxford, South Parks Road Oxford, OX1 3QY, UK. Correspondence and requests for materials should be addressed to C.H. (email: chg@ceh.ac.uk)

Fossil fuel burning is increasing atmospheric carbon dioxide ($CO_2$) concentrations, which both model- and data-based evidence indicates is warming the planet. Approximately 25% of $CO_2$ emissions have been assimilated into terrestrial ecosystems, and whether this continues affects future temperatures. To enable planning for climate change requires robust descriptions of atmospheric $CO_2$ capture by photosynthesis (gross primary productivity; GPP) and release by plant (and soil) respiration. The first climate-carbon cycle projection by a global climate model (GCM), HadCM3[1], identified upper canopy leaf-level dark respiration, $R_d$ ($\mu$mol $CO_2$ m$^{-2}$ s$^{-1}$), as a quantity central to predictions of whole-plant respiration. $R_d$ is parameterised at reference leaf-level temperature 25 °C as $R_{d,25}$ ($\mu$mol $CO_2$ m$^{-2}$ s$^{-1}$). In the HadCM3 simulation[1], $R_{d,25}$ is assumed to be a proportion of maximum carboxylation rate of Rubisco at 25 °C ($V_{cmax,25}$ ($\mu$mol $CO_2$ m$^{-2}$ s$^{-1}$)), itself dependent on mass-based leaf nitrogen concentration, $n_l$ (kg N (kg C)$^{-1}$). At different leaf-level temperatures, $R_d$ follows a $Q_{10} = 2.0$ response, thus doubling over 10 °C intervals, although in newer simulations $R_d$ is suppressed at very high (and low) temperatures[2] (Methods), and with impact of this assessed for the tropics[3]. Under business-as-usual emissions, by year 2100 in that first simulation[1], total ecosystem respiratory $CO_2$ release overtook GPP, changing the land surface to a source of $CO_2$ to the atmosphere. The land surface model in those studies, now named JULES (Joint UK Land Environmental Simulator; Methods and Table 1)[4], continues operation in Hadley Centre GCMs.

Better understanding of plant respiration has become available. Characterisation of $R_d$ in past studies[1, 4] was based on the best available $V_{cmax,25}$-$n_l$ and $R_{d,25}$-$V_{cmax,25}$ parameterisations obtainable at the time (Methods). Recently geographically comprehensive field surveys of $R_d$ and its temperature dependence have become available, including multi-species comparisons. These new datasets include revised estimates of $R_{d,25}$ (GlobResp data)[5], responses to temperature in the short-term[6], and longer-term acclimation-type effects[7–9]. Now required is assessment of how these datasets revise models of the global carbon cycle.

The GlobResp dataset[5] is of upper canopy leaf $R_{d,25}$ from ~ 100 sites distributed around the globe, across several biomes and many species (Methods). GlobResp provides $R_{d,25}$ parameterisation that scales linearly with leaf nitrogen concentration, $n_{l,a}$ (gN (m$^2$ leaf)$^{-1}$), via parameter $r_1$ ($\mu$mol $CO_2$ m$^{-2}$ s$^{-1}$ (gN (m$^2$ leaf)$^{-1}$)$^{-1}$) (along with offset parameter $r_0$ ($\mu$mol $CO_2$ m$^{-2}$ s$^{-1}$)), and in a plant functional type (PFT)-dependent manner. Higher $n_{l,a}$ values increase $R_{d,25}$. A recent compilation[6] of 673 high-temporal resolution, short-term instantaneous responses of leaf $R_d$ to upper-canopy leaf temperature $T_l$ (°C), again from across the globe (Methods), show convergence in leaf $R_d$-$T_l$ response across biomes and PFTs. Analysis of the dataset[6] reveals a single empirical response curve, as an exponential-of-quadratic, fits well and with

two coefficients $b$ and $c$ that gives an effective $Q_{10}$ continuously declining with increasing $T_l$. This is different from earlier observations[10] and models[11, 12]. Across the range of leaf temperatures in nature, application of this response[6] does not predict a decrease in $R_d$ for high $T_l$; we refer to this short-term response as the $b,c$ temperature formulation.

The GlobResp dataset[5] additionally shows leaf $R_{d,25}$ as highest at cold high latitudes, and lowest in warm tropical environments, consistent with acclimation adjustments when plants experience sustained differences in growth temperature[7–9]. Recent modelling studies[13, 14] include thermal acclimation of GPP (via shifts in temperatures at which peak rates of carboxylation ($V_{cmax}$) and electron transport ($J_{max}$) occur[15]), and $R_d$ via mean air temperature of preceding 10 days[16]. The latter study[16] uses data on $R_d$ from juveniles of 19 plant species grown under hydroponic and controlled environment conditions[17]; GlobResp, however, is a dataset roughly 50 times larger and based on mature plants in the field across global climate gradients. Retaining that respiration acclimates to mean air temperature of the preceding 10 days[17] ($T_G$ (°C)), GlobResp implies the most robust procedure to account for thermal acclimation is a linear, $T_G$-dependent perturbation of $R_{d,25}$ (through parameter $r_2$), decreasing by 0.0402 $\mu$mol $CO_2$ m$^{-2}$ s$^{-1}$ °C$^{-1}$ as $T_G$ increases. As timescales down to just 10 days influence $R_{d,25}$, then by some definitions this acclimation includes, implicitly, longer-term evolutionary adaptation effects.

The combination of $R_{d,25} = r_0 + r_1 n_{l,a} - r_2 T_G$ description[5] and $b, c$ formulation[6] for $T_l$, gives upper-canopy leaf-level respiration as:

$$R_d = \left[ r_o + r_1 n_{l,a} - r_2 T_G \right] \times e^{\left[ b(T_l - 25) + c\left( T_l^2 - 25^2 \right) \right]} \qquad (1)$$

with values[6] of $b = 0.1012$ °C$^{-1}$ and $c = -0.0005$ °C$^{-2}$. We now implement this description of $R_d$ in to the JULES large-scale land model[4]. Linear mixed-effects models for the GlobResp dataset show for four PFTs presently in the JULES model, particular parameters (Table 2) capture much species variation across diverse sites. PFT-dependent $n_{l,a}$ are from the TRY database[18]. Our overall finding is that assimilating the comprehensive GlobResp dataset with the JULES terrestrial ecosystem model yields plant respiration rates that are considerably larger than current estimates. The relative importance of contributions (Methods) to revised $R_{d,25}$ values are broad changes to overall baseline having most influence (via parameters $r_0$, $r_1$ and $r_2$ considered together), followed by the specific acclimation dependency and then the relationship with leaf nitrogen.

## Results

**Numerical simulations.** Figure 1 presents implications of new $R_d$ components of Eq. (1). Figure 1a shows for broadleaf trees significant increases across all temperatures in respiration compared

---

**Table 1 Standard JULES parameters used and implications for $R_{d,25}$ calculation**

| Variable | Name in JULES model, or derived quantity, (and units) | Broadleaf tree | Needleleaf tree | Shrubs | C$_3$ grass | C$_4$ grass |
|---|---|---|---|---|---|---|
| $n_{l0}$ | NL0 (kg N (kg C)$^{-1}$) | 0.0369 | 0.0235 | 0.0349 | 0.0480 | 0.0238 |
| $n_e$ | NEFFC3 or NEFFC4 (mol $CO_2$ m$^{-2}$ s$^{-1}$ kg C (kg N)$^{-1}$) | 0.0008 | 0.0008 | 0.0008 | 0.0008 | 0.0004 |
| $V_{cmax,25}$ | From parameters above ($\mu$mol $CO_2$ m$^{-2}$ s$^{-1}$) | 36.8 | 26.4 | 48.0 | 58.4 | 24.0 |
| $f_{dr}$ | FDC3 or FDC4 (Dimensionless) | 0.015 | 0.015 | 0.015 | 0.015 | 0.025 |
| $R_{d,25}$ | From Eqns above ($\mu$mol $CO_2$ m$^{-2}$ s$^{-1}$), but before division by the constraints in denominator [(1 + exp(0.3(13.0-$T_l$))) × (1 + exp(0.3($T_l$-36.0)))] | 0.4428 | 0.282 | 0.4188 | 0.576 | 0.238 |
| Final $R_{d,25}$ | With suppressing constraints from denominator calculated at $T_l = 25.0$ | 0.4157 | 0.2647 | 0.3932 | 0.5407 | 0.2234 |

The standard parameters (Methods) used in the JULES model to calculate $R_{d,25}$ and for each plant functional type (PFT). However, the $n_{l0}$ values use the 50-percentile numbers of the TRY database

**Table 2 Parameter values used in Equation 1**

| Regression coefficient (and units) | Broadleaf trees (BT) | Needleleaf trees (NT) | Shrubs (S) | $C_3$ grasses | $C_4$ grasses |
|---|---|---|---|---|---|
| $r_0$ ($\mu$mol $CO_2$ m$^{-2}$ s$^{-1}$) | $1.7560 \pm 0.2180$ | $1.4995 \pm 0.1793$ | $2.0749 \pm 0.0774$ | $2.1956 \pm 0.1408$ | n/a |
| $r_1$ ($\mu$mol $CO_2$ m$^{-2}$ s$^{-1}$ (gN (m$^2$ leaf)$^{-1}$)$^{-1}$) | $0.2061 \pm 0.0023$ | $0.2061 \pm 0.0023$ | $0.2061 \pm 0.0023$ | $0.2061 \pm 0.0023$ | n/a |
| $r_2$ ($\mu$mol $CO_2$ m$^{-2}$ s$^{-1}$ (°C)$^{-1}$) | $0.0402 \pm 0.0096$ | $0.0402 \pm 0.0096$ | $0.0402 \pm 0.0096$ | $0.0402 \pm 0.0096$ | n/a |
| $n_{l0}$ (kg N (kg C)$^{-1}$) | 0.0369 | 0.0235 | 0.0349 | 0.0480 | 0.0238 |
| $\sigma_l$ (kg C (m$^2$ leaf)$^{-1}$) | 0.0506 | 0.112 | 0.0512 | 0.0248 | 0.0656 |
| $n_{l,a}$ (gN (m$^2$ leaf)$^{-1}$) from $n_{l,0} \times \sigma_l \times 10^3$ | 1.867 | 2.632 | 1.787 | 1.190 | n/a |
| $R_{d,25}$ with $T_G = 25$°C ($\mu$mol $CO_2$ m$^{-2}$ s$^{-1}$) | 1.136 | 1.037 | 1.438 | 1.436 | n/a |

Parameters $r_0$, $r_1$ and $r_2$, which define $R_{d,25}$[5] and for each plant functional type (PFT). Intermediate rows provide values for calculation of $n_{l,a}$. Values of nitrogen content prescribed to Joint UK Land Environmental Simulator (JULES) ($n_{l0}$) and specific leaf density ($\sigma_l$), used to calculate nitrogen content in area-based units, as $n_{l,a}$, are the 50-percentiles across the TRY database for the PFTs. The last row shows values of $R_{d,25}$ calculated using Eq. (1), assuming $T_l = T_G = 25$ °C. The GlobResp database contains global patterns in upper canopy leaf-level respiration, $R_d$, of BTs, NTs, Ss and $C_3$ grasses. Comparable data for $C_4$ grasses remains lacking (n/a), and hence standard JULES values for $R_d$ are used. Uncertainty bounds are ± one standard errors

to standard JULES, when using the new $R_{d,25}$ values ($T_G = 25$ °C) and either $Q_{10} = 2$ or the $b,c$ $T_l$ response. Figure 1b shows the four PFT responses to $T_l$, with revised $R_{d,25}$ values, $T_G$ again 25 °C, and $b,c$ formulation. Figure 1c illustrates strong $R_d$ differences of Eq. (1) between acclimation temperatures $T_G = 15$, 25 and 35 °C (for broadleaf trees). In Fig. 1d, the orange curve is the same $R_d$−$T_l$ response ($T_G = 25$ °C) as in Fig. 1c. However, the red curve sets acclimation temperature equal to instantaneous temperature i.e. $T_G = T_l$. This sensitivity test recognises that although acclimation growth temperature, $T_G$, is determined over longer 10 day periods, higher $T_G$ values will be geographically where $T_l$ is higher and vice versa. This dampens $R_d$ variation in $T_l$. Curve dashed for extremely rare temperatures $T_G > 35$ °C.

JULES scales $R_d$ to canopy-level respiration, $R_{d,c}$ ($\mu$mol $CO_2$ m$^{-2}$ s$^{-1}$). It can calculate $CO_2$ exchange at each canopy level[19], including dependence on vertical decline of leaf nitrogen[19] and differentiation of direct and diffuse radiation[20]. However, data are unavailable for how well Eq. (1) performs at lower canopy levels, even if nitrogen concentration and temperatures are known. Given this, we use a simple big-leaf exponential decline in leaf respiration throughout the canopy, decay co-efficient $k = 0.5$ and dependent on leaf area index (LAI). Implicit is that canopy nitrogen and light levels decay identically, affecting respiration and photosynthesis. A 30% light inhibition of non-photorespiratory mitochondrial $CO_2$ release[21] is included for light above ~ 2 W m$^{-2}$. $R_{d,c}$ is also reduced by any soil moisture stress[4]. Other respiratory components[4] include maintenance respiration of stems and roots. These are modelled as larger for canopies with high LAI, higher nitrogen concentrations and also described as scaled in $R_{d,c}$. Combining stem and root respiration with $R_{d,c}$ gives whole-plant maintenance respiration, $R_{pm}$ (Methods). JULES calculates growth respiration $R_{pg}$ as linearly increasing (via co-efficient $r_g$) with plant GPP minus $R_{pm}$. Combining $R_{pm}$ and $R_{pg}$ gives whole-plant respiration, $R_p$ ($\mu$mol $CO_2$ m$^{-2}$ s$^{-1}$). Hence changes to $R_d$ description influence all respiration components that add together to give $R_p$.

Spatial gridded calculations enable geographical implications of revised $R_d$ description to be determined. JULES is first forced by meteorological conditions near to pre-industrial state, using UK Climate Research Unit data and atmospheric $CO_2$ concentration of 280 ppm. To understand the implications of each part of our new $R_d$ description and as given by Eqn. (1), we add sequentially each component to understand its relative importance. Four sets of simulations are as follows. The first is called Standard—this is standard default JULES (although with TRY-based $n_{l0}$ and $\sigma_l$ values, as in all simulations; $Q_{10}$ response but high/low temperature suppression—see Methods). The second is called New_$R_{d,25}$—this is new baseline alone, hence new $R_{d,25}$ values[5], plus $Q_{10} = 2.0$ (no suppression) $T_l$ response and fixed $T_G = 25$ °C.

The third is called New_$R_{d,25}$_$b,c$ —this is new baseline and instantaneous response i.e. new $R_{d,25}$ values and $T_l$ response with $b,c$ formulation, but still $T_G = 25$ °C. The fourth is called New_$R_{d,25}$_$b,c$_acclim—this is all factors including acclimation, i.e., new $R_{d,25}$ values, $b,c$ formulation and acclimation via variation in $T_G$. The fourth simulations are therefore for the full Eq. (1)[5, 6]. Figure 2a–c shows how each of these new components of our respiration function uniquely influences whole-plant respiration globally for pre-industrial climate forcings. Figure 2a shows annual gridbox mean $R_p$ (weighting PFTs by fractional cover) for New_$R_{d,25}$ minus Standard simulations. This shows that altered baseline through the new $R_{d,25}$ (and removed suppression) causes large $R_p$ increases, including at mid-Northern latitudes and especially for the tropics. Figure 2b shows New_$R_{d,25}$_$b,c$ minus New_$R_{d,25}$, illustrating the implications of the new instantaneous temperature description. The $b,c$ formulation suppresses respiration in mid-latitudes but enhances for the tropics, although changes are smaller than Fig. 2a. Figure 2c presents New_$R_{d,25}$_$b,c$_acclim minus New_$R_{d,25}$_$b,c$, showing acclimation introduction generally increases predicted pre-industrial $R_p$, except in the tropics where acclimation to higher temperatures lowers respiration.

To estimate anthropogenically-induced climate change, changes in near-surface meteorological conditions use the Integrated Model Of Global Effects of climatic aNomalies (IMOGEN) pattern-scaling system[22, 23] (Methods) responding to altered atmospheric greenhouse gas (GHG) concentrations. Patterns are calibrated against 34 GCMs of the Coupled Model Intercomparison Project Phase 5 (CMIP5)[24], with identical method to that originally undertaken for the HadCM3 GCM[22]. We use known historical, then future GHG concentrations of the RCP8.5 business-as-usual representative concentration pathway[25]. Figure 2d shows, for the most complete updated model, New_$R_{d,25}$_$b,c$_acclim, that historical climate change increases $R_p$ in most locations and especially tropics, despite acclimation dampening stimulatory warming effects. Figure 2e presents calculations between years 2015 and 2050, showing $R_d$ with similar changes to recent past in Fig. 2d. Figure 2f presents year 2015 absolute $R_p$ values for New_$R_{d,25}$_$b,c$_acclim case.

Figure 3 presents model output for single illustrative locations and in year 2015. For our four simulations, presented are respiration components ($R_d$, $R_{d,c}$ and $R_p$), plus GPP and NPP. We chose seven sites across South America, a temperate grassland (London) and boreal region shrubs (Siberia). We select multiple South America sites (Methods), as these are some of the few where measurements are available of all respiration components. In general, new $R_{d,25}$ values, whether also with or without adjustment by the $b,c$ formulation and acclimation, give marked increases in predicted respiration. Transition to whole canopy

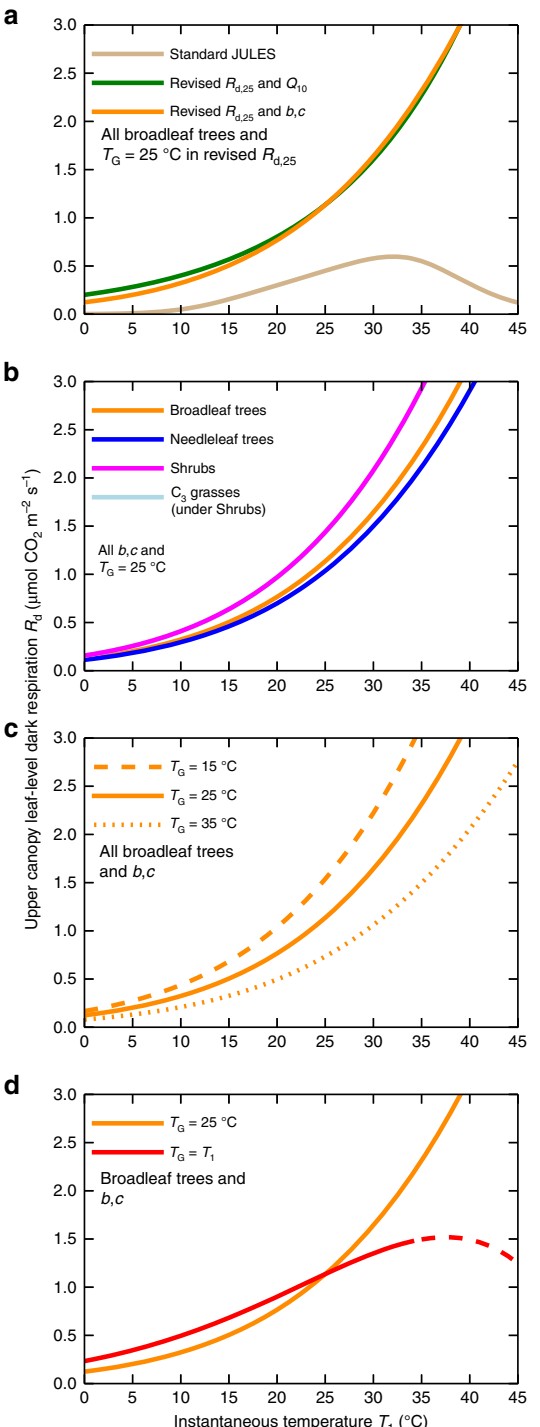

**Fig. 1** Upper canopy leaf-level dark respiration. **a** Standard Joint UK Land Environmental Simulator (JULES) model (with TRY-based $\sigma_l$ and $n_{l0}$ values and $Q_{10}$ response modulated at high and low temperatures (Methods)). Also revised $R_{d,25}$ ($T_G = 25\,°C$) with both $Q_{10} = 2.0$ and $b,c$ responses to $T_l$. **b** New $R_{d,25}$ and $b,c$ response to $T_l$, for other PFTs. $C_3$ grasses near identical curve to Shrubs. **c** New $R_{d,25}$ and $b,c$ formulation for broadleaf trees, but alternative acclimation temperatures. **d** New $R_{d,25}$ and $T_l$ as $b,c$ again broadleaf trees, for both $T_G = 25\,°C$ and $T_G = T_l$. Orange curve common all panels. Light inhibition not included in responses

($R_{d,c}$) and whole plant ($R_p$) respiration illustrates how our leaf level changes propagate to these aggregated fluxes. Uncertainty bounds[5] on $r_0$, $r_1$ and $r_2$ are propagated through the JULES model (Methods) to give uncertainty on $R_{d,c}$ and $R_p$ as shown in Fig. 3,

while measurement uncertainty is from the literature describing each site. For South American sites, and with our choice of big-leaf approximation, our changes reproduce whole-canopy respiration $R_{d,c}$ better (i.e., model and data uncertainty bounds overlap, and better than the default Standard JULES configuration), and in some instances also $R_p$. More specifically, we define the JULES model as having improved performance when the Standard simulation estimate of $R_p$ lies outside the data-based bounds on whole-plant respiration, but simulations New_$R_{d,25}$_$b$,$c$_acclim then fall within those bounds. This happens for the sites at Manaus, Tambopata, Iquitos (dataset a), and Guarayos (dataset a). However, when subtracting $R_p$ from GPP estimates, NPP values are generally too small. We note that observations of nitrogen at different canopy positions from tropical tree species suggest an effective decay co-efficient $k$ with value nearer to 0.2 than 0.5[26]. Using this to scale, and with Eqn (1) still used for upper canopy levels, gives exceptionally large $R_p$ values and unsustainable negative NPP.

Figure 4 shows global time-evolving changes, since pre-industrial times, in total whole-plant respiration, $\Delta R_p$ (GtC yr$^{-1}$) and for our four RCP8.5 scenario simulations. Annotated are pre-industrial and 2015 absolute $R_p$ estimates. Replacement of standard JULES with GlobResp-based[5] $R_{d,25}$ values (still $Q_{10} = 2$, although with no high or low temperature suppression) approximately doubles both pre-industrial respiration estimates (as marked) and the projected changes in $\Delta R_p$ under climate change. Replacing $Q_{10}$ with $b,c$ formulation[6] causes slight global changes. Thermal acclimation increases $R_p$ slightly for pre-industrial but decreases evolving $\Delta R_p$, i.e., comparing simulations New_$R_{d,25}$_$b$,$c$_acclim and New_$R_{d,25}$_$b$,$c$. The stimulatory effect of acclimation arises from the higher predicted rates in globally widespread biomes where $T_g < 25\,°C$, but then dampens responses of such sites to future warming. Our new global $R_p$ values (80.4 GtC yr$^{-1}$ in 2015 for New_$R_{d,25}$_$b$,$c$_acclim simulations) are higher than other estimates for contemporary periods. One[27] global GPP estimate is 119.6 GtC yr$^{-1}$, and balancing soil plus plant respirations will be similar magnitude i.e. together they are also of order 120 GtC yr$^{-1}$. With soil respiration equivalent in size to $R_p$, this suggests plant respiration of order 60 GtC yr$^{-1}$. Our analysis implies global $R_p$ values could be ~30% higher than that value. However, this estimate is not just a consequence of the entrainment of GlobResp data in to the JULES model, but also the scalings within it and as illustrated at selected geographical points in Fig. 3.

The GlobResp database[5] is sufficiently comprehensive as to be globally representative for our simulations. Our analysis has implications for other land ecosystem modelling groups. From a survey of ten leading land surface models, six of these simulate leaf respiration with a dependency on nitrogen content (models listed in Methods). In addition as is common to most land surface carbon cycle models (e.g., also for the Lund Potsdam Jena LPJ model (Table 1 of ref. [28])), the JULES system calculates maintenance respiration for other components of roots and stems that are based on their carbon and estimated nitrogen content. This approach follows pioneering research[29] which moved respiration representation on from simply a fraction of GPP, as had been assumed beforehand. We expect the impact of changing the temperature function itself to at least be common among the current generation of models. However this does open research questions as to how Eq. (1) might change at lower positions in canopies, and whether root, stem and growth respiration models require refinement. This is especially because our GlobResp-based changes propagate directly through modelled canopies by JULES (Eqs. 42–25 in ref. [4]). Hence higher upper-canopy $R_d$ values then generate larger rates of whole-plant respiration, $R_p$, than other estimates.

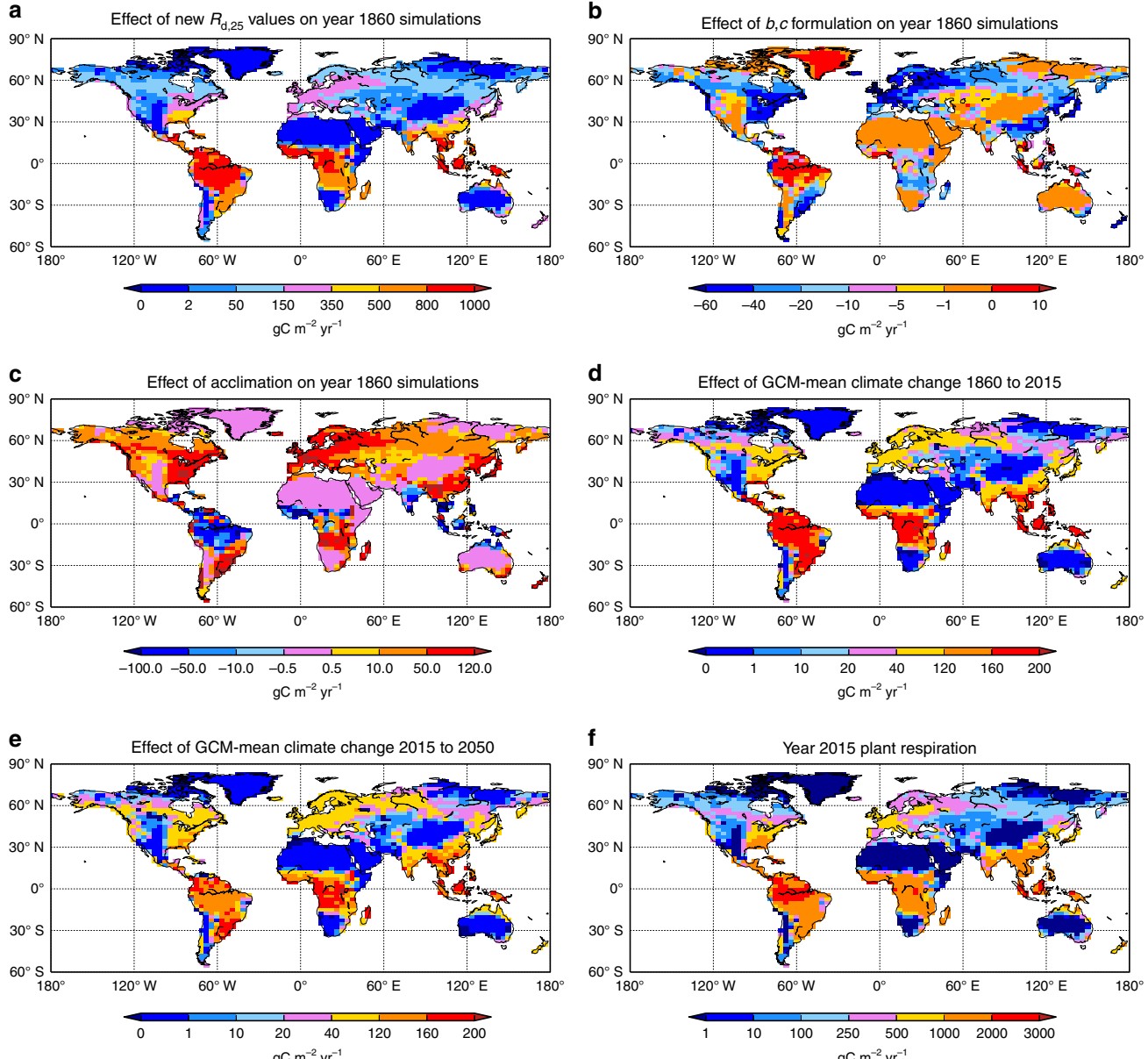

**Fig. 2** Gridbox-mean maps of total plant respiration for new processes and imposed climate change. Changes to $R_p$ as: **a** introduction of new $R_{d,25}$ values: New_$R_{d,25}$ minus Standard; **b** effect of new instantaneous $T_l$ response: New_$R_{d,25}$_b,c minus New_$R_{d,25}$; **c** effect of acclimation: New_$R_{d,25}$_b,c_acclim minus New_$R_{d,25}$_b,c; **d** effect of climate change to present, as year 2015 minus year 1860 and new processes New_$R_{d,25}$_b,c_acclim; **e** similar to **d**, but year 2050 minus year 2015. Panel **f** actual 2015 values of New_$R_{d,25}$_b,c_acclim. Scales different between panels to highlight effects. Units are SI. Panels **d**–**f** are means across the 34 GCMs emulated

Benchmarking tests of modelled respiration fluxes will be important. For instance, the International LAnd Model Benchmarking project (ILAMB)[30] is a comprehensive system collating datasets relevant to land surface functioning and of importance to land surface respiration is the Global Bio-Atmosphere Flux (GBAF)[31] dataset based on extrapolation of eddy-covariance FLUXNET sites. Also available are estimates of global soil respiration[32], which in conjunction with GBAF measurements can return total plant respiration, at least for comparison at night-time periods. Presently, however, without comprehensive measurements of other canopy components, it is difficult to attribute any discrepancies to GlobResp versus lower-canopy, stem, root or growth contributions. Should higher $R_p$ values imply especially low values of NPP, then GPP parameterisation may need reassessment; other analyses suggest current estimates of GPP may be too low[33].

Despite this, in Fig. 5 we perform large-scale comparisons against two Earth Observation-based datasets. These are estimates of NPP from the MODerate-resolution Imaging Spectroradiometer (MODIS) satellite, using the MOD17 algorithm[34, 35], and of GPP from the Model Tree Ensemble (MTE) method[27]. For both datasets, we evaluate mean NPP and GPP values depending on location, and mapping these on to local dominant biomes based on the World Wildlife Fund (WWF) ecoregion classifications[36] (Methods). These data-based estimates locally represent mean NPP and GPP, and so for parity we compare against modelled gridbox mean JULES calculations of the equivalent quantities. That is, we use areal weighting of the five PFT types in JULES for each position. To keep similarities with the WWF categories, we plot in Fig. 5 total annual NPP and GPP for both data and JULES, integrated over areas for the named biomes as marked. Presented

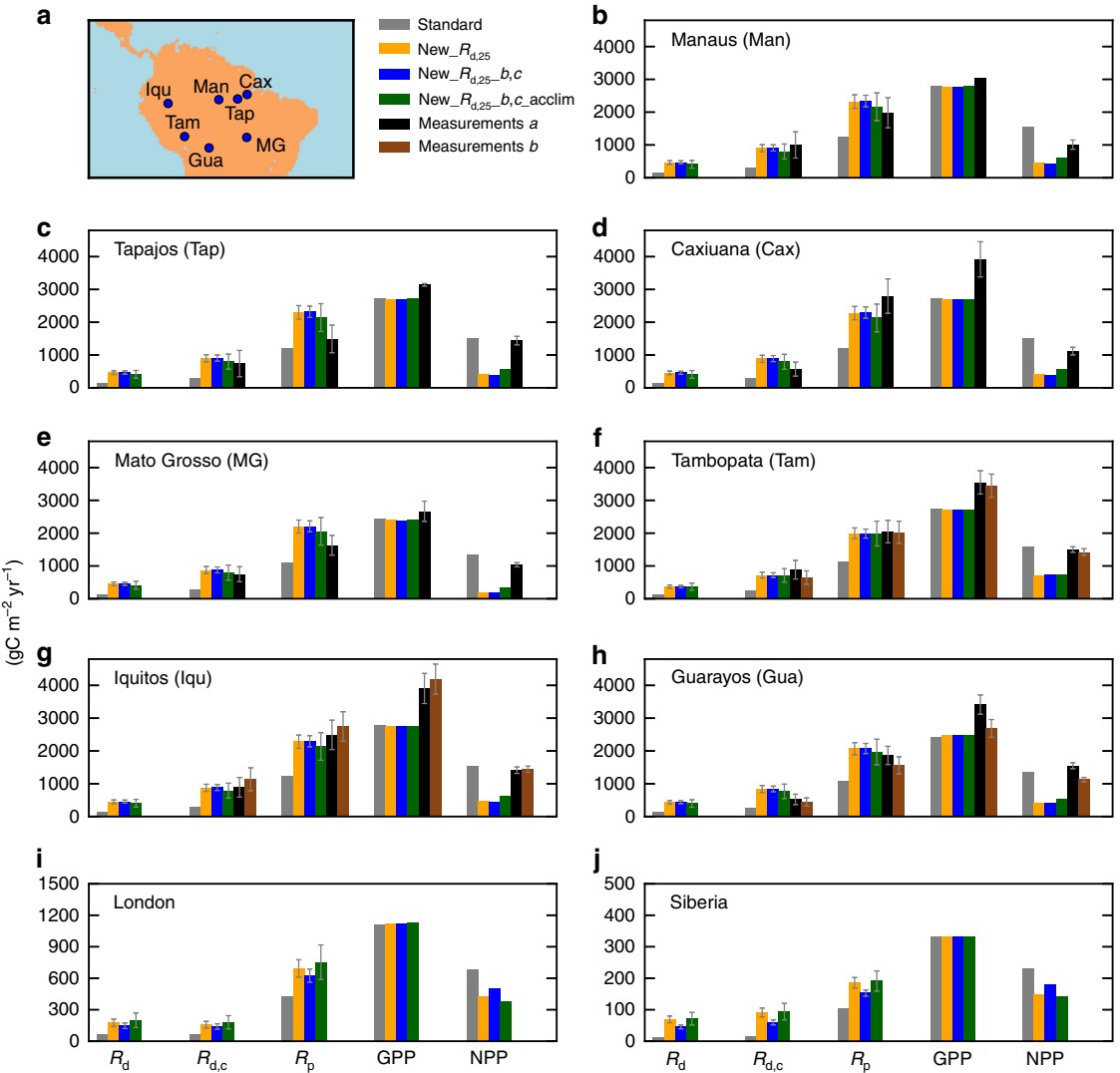

**Fig. 3** Respiration and primary productivities (gross as GPP and net as NPP) at selected locations during modelled year 2015. Seven locations (details in Methods) **a** are South American **b**–**h**, along with **i** for gridbox containing London, UK, and **j** is in Siberia, Russia (Lat 70 N, Lon 82.5 E). Shown for dominant plant functional type (PFT) at each site, left to right, for each histogram cluster: upper canopy leaf-level respiration (with light inhibition) $R_{d}$, whole canopy-level respiration $R_{d,c}$, total plant respiration $R_p$, GPP and NPP. Each histogram cluster are four estimates: Standard, New_$R_{d,25}$, New_$R_{d,25}$_$b,c$ and New_$R_{d,25}$_$b,c$_acclim. South America sites, 5th (or 6th) column are measurements. Dominant PFTs: **b**–**h** BTs, **i** grasses, **j** shrubs. Uncertainty bounds of ± one s.d. are presented which for model estimates are from data-based upper canopy leaf-level uncertainty estimates, subsequently propagated through the model. For measurements, these bounds are taken from the literature

are Standard and New_$R_{d,25}$_$b,c$_acclim simulations. Calculations with New_$R_{d,25}$ and New_$R_{d,25}$_$b,c$ model format are very similar to New_$R_{d,25}$_$b,c$_acclim and so not shown. As expected, in all cases, introduction of GlobResp-based respiration estimates results in much lower modelled NPP values. Furthermore for New_$R_{d,25}$_$b,c$_acclim simulations and all eight biomes, these are significantly less than MODIS-based measurements. The two sets of simulations have similar GPP estimates, illustrating weak indirect couplings in the JULES model between respiration changes and influence (e.g., via hydrological cycle) on gross primary productivity. It is noted in Fig. 5b that JULES model estimates of GPP are similar to the MTE-based data for tropical forests and tropical savannahs. Uncertainty bounds on data adopt the global literature values of ± 15% for NPP[37] and ± 7% for GPP[38]. These are the small horizontal black bars, shown only on New_$R_{d,25}$_$b,c$_acclim red points.

In Fig. 6, we add geographical information to our global data estimates of NPP and GPP, and for corresponding JULES

simulations with all effects, i.e., New_$R_{d,25}$_$b,c$_acclim (expanding on the red symbols of Fig. 5). Figure 6a is JULES NPP estimates divided by MOD17-based NPP estimates (and multiplied by 100 to give percentage). In general modelled NPP with new plant respiration description, is smaller than MOD17 NPP across the geographical points. For some points it can give unsustainable negative modelled NPP values. For GPP, the situation is slightly less clear. Figure 6b is JULES GPP estimates divided by MTE-based GPP values, again as percentage. For many points, the JULES model is also underestimating GPP, and this includes much of the Amazon region. However, for the tropics, a few modelled GPP values are actually higher than data. This offers an explanation as to why GPP appears underestimated in some tropical points of Fig. 3, yet for the average across Tropical Forest (TF), JULES performs well (Fig. 5b). Figure 6b also shows that modelled GPP is usually too low outside of the tropics. This is why, when combined with the enhanced respiration of New_$R_{d,25}$_$b,c$_acclim formulation, this can lead to very low or

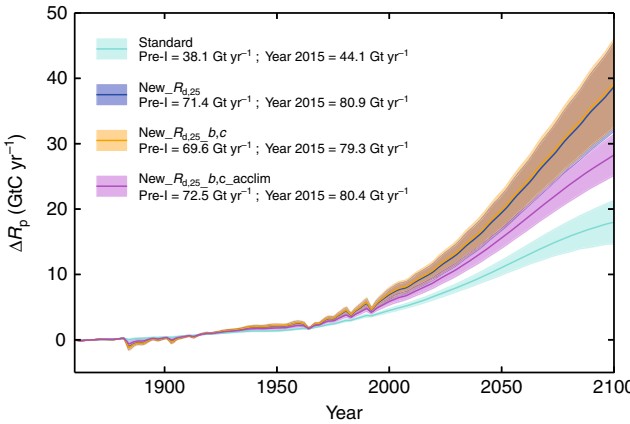

**Fig. 4** Time series of change in areally-averaged global respiration. Presented are time-evolving model estimates of change in total whole-plant respiration, $\Delta R_p$. The colours of turquoise, blue, yellow and magenta are Standard, New_$R_{d,25}$, New_$R_{d,25}$_b,c and New_$R_{d,25}$_b,c_acclim respectively. Where yellow and blue projections overlap, the colour is brown. The spread corresponds to the different projections of climate drivers, based on the 34 Global Circulation Models (GCMs) emulated in the Integrated Model Of Global Effects of climatic aNomalies (IMOGEN) modelling system and for RCP8.5 scenario. The continuous lines are the mean, and the spread as ± two s.d. which broadly covers inter-GCM spread. Pre-industrial (marked Pre-I) and year 2015 model mean absolute estimates of $R_p$ are as annotations

even unsustainable negative NPP. Figure 6c shows the dominant WWF-defined biomes for each location.

## Discussion

Inversion studies suggest roughly 25% of $CO_2$ emissions are presently assimilated by the land surface[39]. Hence net ecosystem productivity (NEP) is ~ 2.5 GtC yr$^{-1}$, implying a small difference between GPP and total ecosystem respiration (whole-plant plus soil) fluxes. Here we have entrained the GlobResp dataset[5] of upper-canopy respiration with a well-established land surface model JULES which aggregates $R_d$ through to whole-plant respiration. This implies higher whole-plant respiration, and therefore may need to be balanced by either higher GPP values[33] or the multiplicative dependence of other components on $R_d$ is too large. As global land-atmosphere $CO_2$ fluxes are a small difference between large fluxes, future terrestrial ecosystem respiration responses to warming can therefore influence the natural ability to offset $CO_2$ emissions. This is particularly important as land warmings are projected to be higher than global mean rise[40]. The recent pause in growth rate of atmospheric carbon dioxide has been linked to the warming hiatus suppressing respiration whilst $CO_2$ fertilisation continues[41]. If future increases in respiration overtake any thermal or $CO_2$-ecosystem fertilisation, lower NPP values in the most extreme instances could force biome changes[1]; this will require operation of the interactive vegetation component of land surface models to assess (Methods). Equivalent global respiration measurement campaigns to GlobResp, but for other canopy components, will aid our understanding of the likelihood of respiration-induced biome changes. Such additional data will enable more rigorous benchmarking of different terrestrial model configurations of within-canopy respiration fluxes. Full mechanistic models, which can still be tested against GlobResp data, ultimately may allow further advances on empirical-based descriptions of respiration. However, availability of these remains a long way from routine

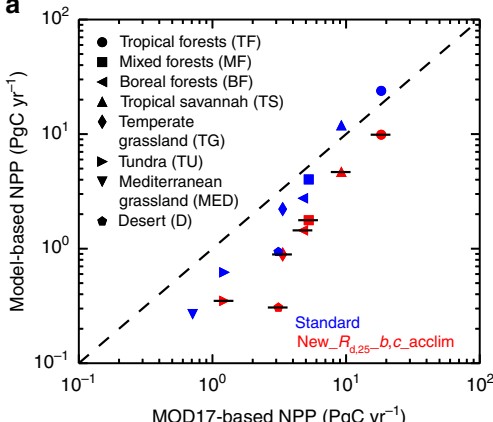

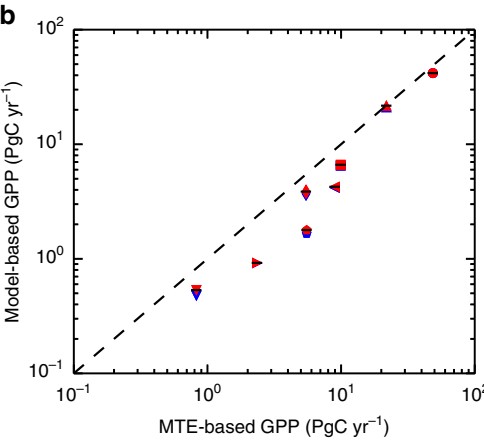

**Fig. 5** Data- and model-based global estimates of net primary productivity and gross primary productivity for different biomes. **a** Global measurements of total annual mean net primary productivity (NPP), average for years 2000–2011, and using Earth-observed MODerate-resolution Imaging Spectroradiometer (MODIS) measurements. Values are spatially aggregated for different World Wildlife Fund (WWF) biome classifications. The dominant biome type at each location is linked to NPP with the MOD17 algorithm applied to MODIS values (horizontal axis). Similarly gridbox-mean JULES estimates of NPP are multiplied by gridbox area, and combined for each WWF biome (vertical axis). This is dependent upon which WWF biome is dominant for the grid location. Note logarithmic axes. JULES NPP estimates are slightly negative for Mediterranean grasslands and so off axes. **b** Similar calculation for gross primary productivity (GPP), with measurements from the Model Tree Ensemble (MTE) algorithm. Both panels, model values presented in blue for standard JULES version (i.e., Standard simulation), and in red for new $R_{d,25}$ values with b,c temperature response and acclimation (i.e., New_$R_{d,25}$_b,c_acclim simulation). For GPP, differences are small between two model forms, with red symbols overlapping blue symbols. Uncertainty bounds on NPP and GPP data are the small black horizontal bars ( ± one s.d.), shown for red symbols only. All calculations include only land points with less than 50% agriculture

usage, yet alone in large-scale climate models. This is an issue recently discussed in depth for the b,c instantaneous temperature response formulation[42, 43], and where that exchange in the literature has relevance to more general respiration modelling.

## Methods

**Datasets**. Two recently reported global datasets underpin Eq. (1). GlobResp describes patterns of temperature-normalised leaf respiration and associated leaf

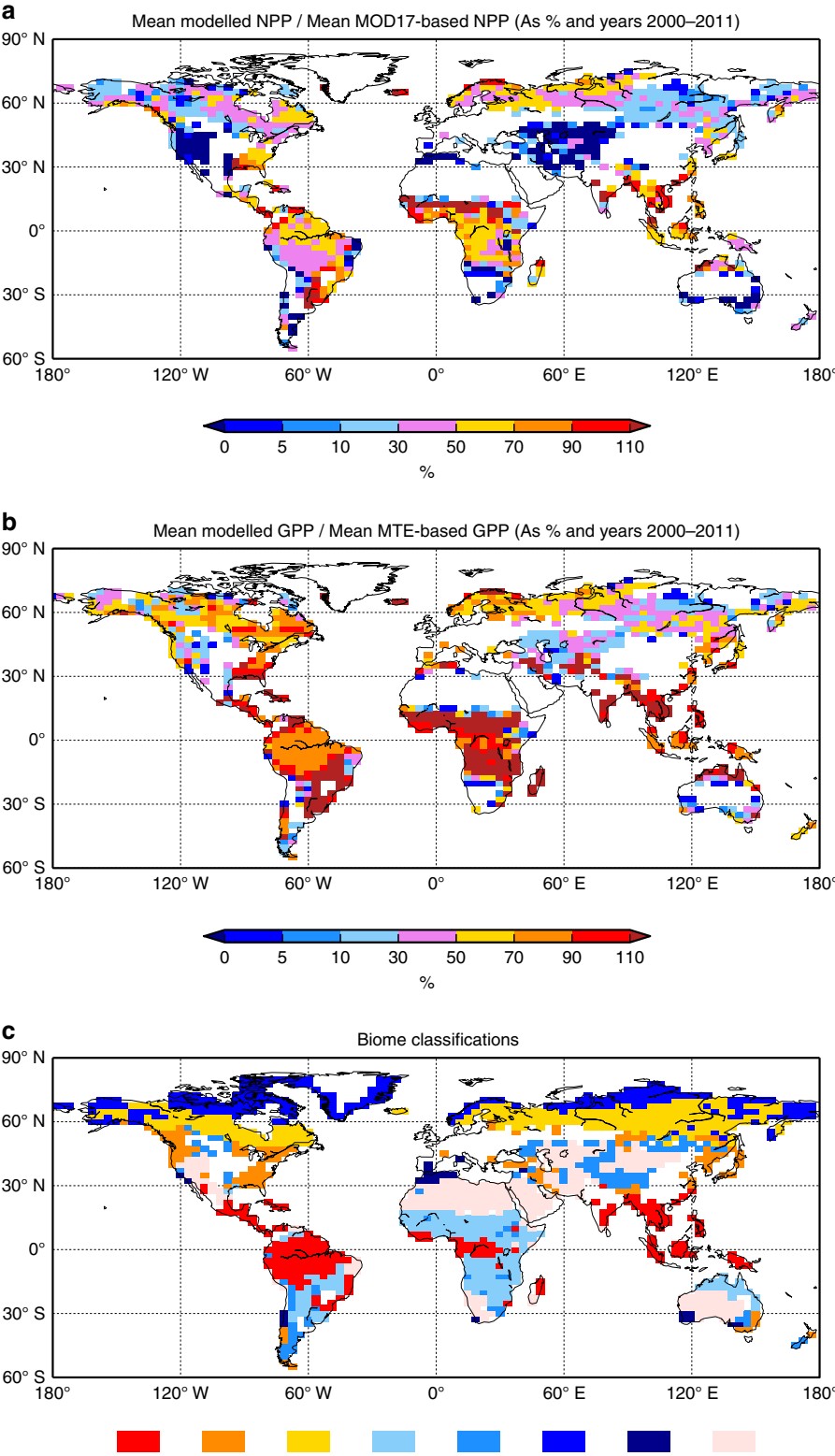

**Fig. 6** Data- and model-based maps of comparison of net primary productivity and gross primary productivity for different biomes. **a** Map of JULES estimates of annual NPP, average for year 2000-2011 divided by MODIS NPP algorithm (MOD17) estimates for the same period. Values multiplied by one hundred to express as percentage. Land points excluded are those with >50% agriculture, and also where values are very small (if absolute value of JULES or MODIS NPP is less than 1 gC m$^{-2}$ yr$^{-1}$). **b** Similar to (**a**) but for GPP, and data based on upscaled FLUXNET GPP from the MTE algorithm. Again, land points excluded are those with > 50% agriculture, and those with small values (if value of JULES or MTE-based GPP is less than 1 gC m$^{-2}$ yr$^{-1}$). Panel **c** is map of dominant biome, and labels identical to Fig. 5

traits[5]. Respiration rates of sun-exposed leaves were measured for ~ 900 species, from 100 globally distributed sites covering from Arctic to the tropics. For each species, leaf respiration in darkness was measured during the day in situ on attached branches, or using detached branches that had their stems re-cut under water to maintain water flow to leaves. Leaves were dark-adapted for 30 min before each measurement, with respiratory $CO_2$ release being measured using infra-red gas analyzers. Leaves were sampled, dried and analysed for total leaf nitrogen. The database[5] shows that PFTs used by the JULES model capture much species variation across diverse sites. Respiration acclimates[16] to prevailing ambient growth temperature, $T_G$, and responses confirm (for identical temperatures) that cold-grown plants exhibit higher respiration rates than their warm-grown counterparts.[8,9]

The second dataset describes variations in leaf respiration (in darkness) to instantaneous temperature changes[6] based on 673 respiration-temperature curves from 231 species and 18 field sites. Leaves of detached branches of sun-exposed leaves were placed in a temperature-controlled cuvette and allowed to dark-adapt; leaf respiration was measured (using a Licor 6400 gas exchange system) as leaves were heated[6] from 10 to 45 °C, at rate of 1 °C min$^{-1}$. Convergence occurred in short-term temperature responses of $R_d$ across biomes and PFTs; a model describing this $T_l$- dependence is an exponential-of-quadratic.

**JULES modelling framework**. The original JULES $R_d$ description, with $Q_{10} = 2.0$, satisfies either $R_d = R_{d,25} Q_{10}^{0.1(T-25)}$, or, with suppression at high and low temperatures, as (Eq. (18) of ref. [2]) via additional denominator: $R_d = R_{d,25} [Q_{10}^{0.1(T_l-25)}] / [(1 + \exp(0.3(13.0-T_l))) \times (1 + \exp(0.3(T_l-36.0)))]$. This latter form is our Standard JULES simulations, and has similarities to others modelling respiration as linear in GPP, with GPP itself a peaked Arrhenius[44] function of $T_l$. The JULES value of $R_{d,25}$ is linear in the maximum carboxylation rate of Rubisco at 25 °C, $V_{cmax,25}$ ($\mu$mol $CO_2$ m$^{-2}$ s$^{-1}$)[45,46]. Parameter $f_{dr}$ relates $V_{cmax,25}$ to $R_{d,25}$ as $R_{d,25} = f_{dr} \times V_{cmax,25}$, where $V_{cmax,25}$ satisfies[47] $V_{cmax,25} = 10^6 \times n_e \times n_{l0}$. Quantity $n_{l0}$ is the prescribed mass-based PFT leaf nitrogen concentration (kg N (kg C)$^{-1}$) and $n_e$ (mol $CO_2$ m$^{-2}$ s$^{-1}$ kg C (k;gN)$^{-1}$) links $V_{cmax,25}$ to leaf nitrogen concentration. Table 1 shows how these equations and parameters give Standard JULES $R_{d,25}$ values.

The parameters of Eq. (1) are given in Table 2, along with implication of GlobResp-based values[5] for $R_{d,25}$ values, when incorporated in to the JULES model.

The relative importance of contributions to revised $R_{d,25}$ can be assessed from Tables 1 and 2. In general terms, and for broadleaf trees, the new representative $R_{d,25}$ values change from 0.4157 to 1.136 $\mu$mol $CO_2$ m$^{-2}$ s$^{-1}$. From the TRY database[18], with 80% confidence, leaf nitrogen concentrations lie between 62% and 154% of their median value. This gives a range of $0.237 < r_1 n_{l,a} < 0.593$ $\mu$mol $CO_2$ m$^{-2}$ s$^{-1}$. Growth temperature ranges of 5–25 °C give $0.2 < r_2 T_G < 1.0$ $\mu$mol $CO_2$ m$^{-2}$ s$^{-1}$. This simple scale argument suggests a decreasing importance, both in terms of absolute and potential variability, of contributions to new $R_{d,25}$ as new baseline, followed by acclimation and then by leaf nitrogen dependence.

Scaling to full canopy, respiration is modelled as declining exponentially in LAI above each point, $L$, as $R_d \exp(-kL)$ and $k = 0.5$. Hence all-canopy respiration $R_{d,c} = R_d \times [1-\exp(-kL)]/k$. This has modulation of $R_d$ with light inhibition[21], and any low soil moisture constraints[4]. Three additional components of respiration are those of roots, stems and growth. Root and stem respiration[4] are linear in $R_{d,c}$ (and thus $R_d$) and dependent on estimated nitrogen concentrations in each. Canopy, root and stem respiration combine to an overall whole-plant maintenance respiration $R_{p,m}$. Growth respiration, $R_{p,g}$ is assumed to be a fixed fraction of GPP ($\Pi_g$) minus $R_{p,m}$ as $R_{p,g} = r_g \times [\Pi_g - R_{p,m}]$, with coefficient $r_g = 0.25$. Whole plant respiration, $R_p$, is the sum of maintenance $R_{p,m}$ and growth $R_{p,g}$ respiration, as $R_p = R_{p,g} + R_{p,m}$.

JULES has a Dynamic Global Vegetation Model (DGVM) option, named Top-down Representation of Interactive Foliage and Flora Including Dynamics (TRIFFID)[4], which uses calculated NPP (i.e. GPP-$R_p$) to estimate LAI, in turn influencing PFT competition to derive their areal cover. This interactive vegetation model component enables, under major climate change, estimation of potential biome cover changes. To understand the implications of Eq. (1) on $R_p$ without extra feedbacks via varying LAI, for this study we prescribe representative LAI and fractional covers for the PFTs at different geographical positions. This includes a prescribed fractional cover of agriculture, which is broadly representative of the current period, thus over-riding the TRIFFID component. However once additional data to refine within-canopy, root, stem and growth respiration estimates is available, building more confidence in total plant respiration estimates, then the TRIFFID component can be operated to assess future biome change likelihood. In the most general terms, the LAI and fractional cover of a PFT is strongly dependent on calculated NPP. If respiration increases significantly, this will lower NPP, lower LAI and reduce respiration (although it would also lower GPP).

**IMOGEN climate impacts modelling framework**. The IMOGEN modelling system[23] uses pattern-scaling to emulate GCMs. Radiative forcing $Q$ (W m$^{-2}$) is a single metric across greenhouse gases and aerosols that describes their overall influence on perturbed atmospheric energy exchanges. In IMOGEN, changes in $Q$ drive a global energy balance model (EBM), which in turn multiplies patterns of climate change[22]. Such patterns are change in local and monthly meteorological

conditions for each degree of global warming over land, with the latter estimated by the EBM. EBM parameterisation and patterns have been fitted against 34 GCMs in the Coupled Model Intercomparison Project Phase 5 (CMIP5) database[24], including mapping from GCM native grids to 2.5° latitude × 3.75° longitude. Our climate change simulations are for historical, followed by atmospheric GHG concentrations of the RCP8.5 business-as-usual pathway[25].

**Uncertainty analysis and site data**. Uncertainty bounds[5] on $r_0$, $r_1$ and $r_2$ are repeated in Table 2. Naming these standard deviations as $\varepsilon_0$, $\varepsilon_1$ and $\varepsilon_2$, then uncertainty on $R_{d,25}$ for the New_$R_{d,25}$ and New_$R_{d,25}\_b,c$ is $\sqrt{\varepsilon_0^2 + n_{l,a}^2 \varepsilon_1^2}$ whilst that for New_$R_{d,25}\_b,c\_acclim$ is $\sqrt{\varepsilon_0^2 + n_{l,a}^2 \varepsilon_1^2 + T_G^2 \varepsilon_2^2}$. This therefore assumes that each of the three individual uncertainties are independent of each other. Bounds on parameter $b$ and $c$ in Eq. (1) are negligible[6] and so multiplication by the exponent of Eq. (1) gives overall bounds on $R_d$. This uncertainty is then passed through the JULES aggregation scheme in a similar way as that for the absolute dark respiration values, to give bounds on $R_{d,c}$ and $R_p$. Data and its related uncertainty bounds for Fig. 3 is from Manaus and Tapajos[48], Caxiuana[49], Mato Grosso[50], Tambopata[51], Iquitos[52] and Guarayos[53]. Measurements a and b in locations Fig. 3f–h refer to different plots at the same location. At Tambopata, measurement a is for plot TAM05 and measurement b is for plot TAM06[51]; at Iquitos, measurement a is for plot Alp A and measurement b is for plot Alp C[52]; and at Guarayos, measurement a is for plot Kenia-dry and measurement b is for plot Kenia-wet[53].

A review of the dependencies of ten other major land surface models shows that for six of these, upper canopy leaf respiration is dependent on leaf nitrogen content. The dependences are: BETHY is $V_{c,max}$[54]; BIOME3 is $V_{c,max}$[55]; BIOME-BGC is Nitrogen[56]; Century is Nitrogen[57]; CLM is Nitrogen[58]; LPJ is Nitrogen[28]; O-CN is Nitrogen[59]; Orchidee is Empirical[60]; Sheffield-DGVM is Nitrogen[61] and TEM is Empirical[57]. The two models with dependence on $V_{c,max}$ contain an implicit dependence on nitrogen, via assumed $V_{c,max}$-N relationships.

In Fig. 5, we present data and model-based estimates of global NPP and GPP, divided into eight biomes that are in turn based on 13 in the WWF definitions of terrestrial ecoregions[36]. This reduction is by merging tropical, subtropical forests and mangroves into tropical forests; merging temperate mixed-forests and temperate conifers into (extratropical) mixed-forests, and merging temperate grasses, flooded grasses and montane grasses into temperate grassland.

**Data availability**. The GlobResp data is freely available from the TRY Plant Trait Database http://www.try-db.org/TryWeb/Home.php. The JULES model is freely available at http://jules.jchmr.org/. The code changes to the JULES respiration subroutine used in this analysis are available on request from Chris Huntingford (chg@ceh.ac.uk). All JULES model outputs, and for the four factorial experiments, are available for download from the Environmental Information Data Centre. The address is: https://doi.org/10.5285/24489399-5c99-4050-93ee-58ac4b09341a.

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

## Acknowledgements

C.H. acknowledges the NERC CEH National Capability fund. We acknowledge the many climate research centres that contributed GCM outputs in to the Coupled Model Intercomparison Project (CMIP5) database. The support of the Australian Research Council to O.K.A. and P.M. (DP130101252, CE140100008, FT0991448, FT110100457) is acknowledged, as are awards DE-FG02-07ER64456 from the US Department of Energy, Office of Science, Office of Biological and Environmental Research and DEB-1234162 from the U.S. National Science Foundation (NSF) Long-Term Ecological Research Program (to P.B.R.); and National Science Foundation International Polar Year Grant (to K.L.G.). L.M.M. acknowledges the support of the Natural Environment Research Council

(NERC) South American Biomass Burning Analysis (SAMBBA) project grant code NE/J010057/1.

## Author contributions

C.H. and O.K.A. conceived the project. C.H. designed the paper and research methodology. O.K.A., M.A.H., L.M.M., K.J.B., O.S.O., K.R.W., P.B.R., E.E.B., M.C., K.L.G., P.M., M.G.T. and M.H.T. intepreted the *GlobResp* data and its mapping on to the JULES model. A.M., L.M.M., A.B.H., S.S. and K.W. developed the JULES model. L.M.M., A.M. and Y.M. related JULES simulations to other measurements (Fig. 3). All authors contributed to the writing of the paper.

## Additional information

**Competing interests:** The authors declare no competing financial interests.

