## [Peer Review File · Nature Communications]

Reviewers' Comments:

Reviewer #1 (Remarks to the Author)

Plant respiration is far more complex than representations in Earth System Models. The simplification may preclude accurate predictions of C balance of land surface in future. Thermal acclimation of respiration certainly influences base respiration rates and thermal sensitivity of plants across biomes, and there is no reason to suspect that plant respiration will not acclimate to future temperatures. A previous study by another co-author (Atkins et al 2015) provides a framework for incorporating the dataset into terrestrial biosphere models and this paper appears to be the fruition of that effort.

The dataset used to parameterize the new respiration implementation appears to be significantly more robust than any previous dataset. This dataset (Globresp) has been the basis of several important papers in recent years. While there are many processes not included in LSMs, this process is fundamental to forecasting land carbon balance. The implementation of the new parameterization appears to be much needed.

While the dataset used to parameterize the new model is globally extensive whether or not the implementation the new R_p routine in this single model is broadly and globally significant is not demonstrated. It could be - and the authors have altered the paper to include some caveats and list several datasets that could be used to benchmark global models.

They argue that: "Benchmarking tests of modelled respiration fluxes will be important, although presently without comprehensive measurements of other canopy components, it is not possible to attribute any discrepancies to GlobResp versus lower-canopy, stem, root or growth contributions"

While the authors are right that more extensive dataset to estimate parameters IMPLIES an improvement in the model, it does not provide evidence for that the global estimates of respiration have been improved. The authors chose to limit any comparisons with data to a few locations: "Figure 3 shows model output for single illustrative locations.....We select multiple South American sites, as these are of the few where measurements are available of all respiration components."

This could provide some support to their case but a statistical assessment of the modeled respiration against the measurements is not reported - nor are there any uncertainties presented for the data.(from Mahli et al the uncertainty for R_{leaf} is 2.84 Mg C ha⁻¹ year⁻¹ for Tambopata 05 and 2.07 Mg C ha⁻¹ year⁻¹ for Tambopata 06).

After altering one component of the model, there is no reason not to show an evaluation of the model's estimates of total respiration or other fields against one or many of the established benchmarks for LSMs. If the model has improved the leaf respiration then the model should perform adequately against total ecosystem respiration, if it doesn't that would indicate a problem. Leaving out this information just leaves the reading guessing.

I would have expected the paper to flow as follows 1) improved model structure 2) parameterization 3) rigorous validation at a limited number of sites and 4) broad comparison with established benchmarks 5) implications. As the paper stands step 3 is not rigorous, step 4 is absent and so the implications are not effectively demonstrated. The conclusions are therefore highly tempered in this version.

Reviewer #2 (Remarks to the Author)

The article "Implications of improved representations of plant respiration in a changing climate" by Huntingford et al. examines the carbon cycle impact of revising the temperature responses of plant respiration within the JULES land surface model. The authors find that the new representations of basal rate respiration increase global plant respiration compared to the standard JULES model, an effect that is partially offset by adding acclimation responses. This is the second Nature journal I have reviewed this article for. I still feel that this is an important and timely study. However, I have many of the same reservations, which I don't feel were adequately addressed from the previous iteration of the manuscript I reviewed. Unfortunately, the authors were not able to present a response to previous reviews to explain how and why they addressed my previous concerns. My main concern is with the implications of the revised formulas. I feel they are overly broad and should be rephrased. I lay out my reasoning below. To summarize, I am concerned that the main results are very model specific, which is okay, but needs to be clearly stated. I am also not convinced that the revised formulations provide substantial model improvements at the large scales at which the conclusions are drawn. The sparse model-data comparison presented is great, but certainly does not convince me that the formulations are better.

- The main conclusion from the abstract is that revised respiration estimates could be 30% higher than previous estimates. The previous estimates were from a single model (JULES). While this is interesting, it is highly model specific and the effect is likely to differ, possibly in both magnitude and direction if compared to other models. This caveat needs to be explicitly stated throughout, including in the abstract. In my opinion, the more interesting comparisons are to the different revised formulations, which have greater commonalities, rather than the comparison to the standard JULES.

- Following from the previous comment, the main result stems primarily from a change in the basal rate respiration calculation within JULES, which involved a couple of important steps: (1) the switch from a R_d-V_{cmax} to R_d-N relationship and (2) the implementation of GLOBRESP parameterizations. Do the authors have a sense for which of these is driving the effect? The sense I get from reading the manuscript is that the latter effect is dominant, although I would think that former may be quite important, primarily due to the uncertainty in estimating leaf/whole plant nitrogen contents throughout the canopy.

- To address the model specificity problem, the authors could add lines from other land surface models to Figure 1.

- From the model-data comparisons, I am not convinced that the revised formulas work better for simulating plant respiration at large scales, which undermines the conclusions a bit. I am sympathetic with the fact that respiration at large scales are sparse, but even in the few sites examined the results are quite mixed, with some estimates being improved at some sites, but not others and the improvement within sites dependent on the estimate evaluated. I think these comparisons should be given a greater emphasis in the study, particularly over the broad and uncertain conclusions from the global simulations.

- I would like to see a greater discussion of the empirical nature of plant respiration formulations in land surface models such as JULES. While the dataset used for the new parameterizations is large relative to its predecessors, it is quite limited in space and time compared to the data simulated by the global simulations. This is a problem given that the mechanisms underlying the respiration responses to temperature and leaf N are not explicitly simulated, but rather assumed from empirical relationships, which is an issue when extrapolating to larger scales. This study is a clear step forward for improving the empirical relationships, but it should be made clearer that these still suffer from a lack of a mechanistic incorporation of plant respiratory responses to temperature and nitrogen.

We have addressed all reviewer requests, and as described below. Please find our replies in blue and indented.

Reviewers' comments:

Reviewer #1 (Remarks to the Author):

Dear Reviewer 1. First, can we thank you for your time assessing our analysis, and especially as you have reviewed it more than once. Your comments and suggestions have helped the manuscript significantly including new figures, better rigorous statistical analysis and encouraging us to better place the *GlobResp*-based simulations in a general context.

Plant respiration is far more complex than representations in Earth System Models. The simplification may preclude accurate predictions of C balance of land surface in future. Thermal acclimation of respiration certainly influences base respiration rates and thermal sensitivity of plants across biomes, and there is no reason to suspect that plant respiration will not acclimate to future temperatures. A previous study by another co-author (Atkins et al 2015) provides a framework for incorporating the dataset into terrestrial biosphere models and this paper appears to be the fruition of that effort.

Thank you. Yes, that is the intention. The *GlobResp* data is unique, in both its extensiveness of number of samples, and geographical distribution. However from the original data-based Atkin et al (2015) paper, it was not possible to understand the global implications. Here we have addressed that with a global model.

The dataset used to parameterize the new respiration implementation appears to be significantly more robust than any previous dataset. This dataset (Globresp) has been the basis of several important papers in recent years. While there are many processes not included in LSMs, this process is fundamental to forecasting land carbon balance. The implementation of the new parameterization appears to be much needed.

Please see comment above.

While the dataset used to parameterize the new model is globally extensive whether or not the implementation the new R_p routine in this single model is broadly and globally significant is not demonstrated. It could be - and the authors have altered the paper to include some caveats and list several datasets that could be used to benchmark global models.

We first answer the broad significance in terms of other land surface models. However please also see our paper amendments in response to the other detailed and related comments, and that are below this one.

We have reviewed the literature of ten other mainstream land surface models. We find six of these have a direct respiration dependence on nitrogen, whilst two others are via $V_{c,max}$ where there is also an implicit dependence on nitrogen. This suggests our

findings of mapping GlobResp on to the JULES will be representative of other land surface models. We now write in the main paper:

“Our analysis has implications for other ecosystem modelling groups. From a survey of ten leading land surface models, six of these simulate leaf respiration with a dependency on nitrogen content (models listed in Methods).”

In Methods we write: *“A review of the dependencies of ten other major land surface models shows that for six of these, upper canopy leaf respiration is dependent on leaf nitrogen content. The dependences are: BETHY is $V_{c,max}^{52}$; BIOME3 is $V_{c,max}^{53}$; BIOME-BGC is Nitrogen⁵⁴; Century is Nitrogen⁵⁵; CLM is Nitrogen⁵⁶; LPJ is Nitrogen²⁸; O-CN is Nitrogen⁵⁷; Orchidee is Empirical⁵⁸; Sheffield-DGVM is Nitrogen⁵⁹ and TEM is Empirical⁵⁵. The two models with dependence on $V_{c,max}$ contain an implicit dependence on nitrogen, via assumed $V_{c,max}$ -N relationships.”*

- 52 Ziehn, T., Kattge, J., Knorr, W. & Scholze, M. Improving the predictability of global CO₂ assimilation rates under climate change. *Geophysical Research Letters* **38**, doi:10.1029/2011GL047182 (2011).
- 53 Haxeltine, A. & Prentice, I. C. A general model for the light-use efficiency of primary production. *Functional Ecology* **10**, 551-561, doi:10.2307/2390165 (1996).
- 54 White, M. A., P.E., T., S.W., R. & R.R., N. Parameterization and Sensitivity Analysis of the BIOME-BGC Terrestrial Ecosystem Model: Net Primary Production Controls. *Earth Interactions* **4**, 1-85, doi:10.1175/1087-3562(2000)004<0003:pasao>2.0.co;2 (2000).
- 55 Melillo, J. M. *et al.* Global climate-change and terrestrial net primary production. *Nature* **363**, 234-240, doi:10.1038/363234a0 (1993).
- 56 Lawrence, D. M. *et al.* Parameterization Improvements and Functional and Structural Advances in Version 4 of the Community Land Model. *J. Adv. Model. Earth Syst.* **3**, 27, doi:10.1029/2011ms000045 (2011).
- 57 Zaehle, S. & Friend, A. D. Carbon and nitrogen cycle dynamics in the O-CN land surface model: 1. Model description, site-scale evaluation, and sensitivity to parameter estimates. *Global Biogeochemical Cycles* **24**, 13, doi:10.1029/2009gb003521 (2010).
- 58 Krinner, G. *et al.* A dynamic global vegetation model for studies of the coupled atmosphere-biosphere system. *Global Biogeochemical Cycles* **19**, 44, doi:10.1029/2003gb002199 (2005).
- 59 Woodward, F. I. & Lomas, M. R. Vegetation dynamics - simulating responses to climatic change. *Biol. Rev.* **79**, 643-670, doi:10.1017/s1464793103006419 (2004).

They argue that: "Benchmarking tests of modelled respiration fluxes will be important, although presently without comprehensive measurements of other canopy components, it is not possible to attribute any discrepancies to GlobResp versus lower-canopy, stem, root or growth contributions"

In response to this comment, and also the one above and below, to address further “global significance”, we have extended our analysis substantially. We now provide a new Figure 5 that uses Earth Observation measurements of NPP (and GPP) to understand the new respiration simulations, and at the very large scale. We are very careful to not imply all respiration components are now precisely described, although we are now more confident of upper-canopy estimates due to the *GlobResp* database.

Our new Figure 5 and related text gives guidance as to what remain large scale gaps in modelling Net Primary Productivity (NPP) and measurements. This suggests the need for measurement campaigns as comprehensive of *GlobResp* to constrain within canopy, root and stem respiration rates – along with possible further refinement of photosynthesis.

We now write: “.. in Figure 5 we perform large-scale comparisons against two Earth Observation-based datasets. These are estimates of NPP from the MODIS satellite, using the MOD17 algorithm^{36, 37}, and of GPP from the Model Tree Ensemble (MTE) method³⁸. Both of these algorithms evaluate mean NPP and GPP values depending on location, and are mapped on to local dominant biomes in turn based on the World Wildlife Fund (WWF) ecoregion classifications³⁹ (Methods). These data-based estimates, locally, represent mean NPP and GPP, and so for parity we compare against modelled gridbox mean JULES calculations of the equivalent fluxes. That is, we use areal weighting of the five PFT types in JULES for each position. To keep equivalence with the WWF categories, we plot in Figure 5 total annual NPP and GPP for both data and JULES, integrated over areas for the biomes as marked. Presented are ‘standard’ and ‘new $R_{a,25+b,c+acclim}$ ’ simulations. Calculations with ‘new $R_{a,25}$ ’ and ‘new $R_{a,25+b,c}$ ’ model format are very similar to ‘new $R_{a,25+b,c+acclim}$ ’ and so not shown. As expected, in all cases, introduction of *GlobResp*-based respiration estimates results in significantly lower modelled NPP values. Furthermore for ‘new $R_{a,25+b,c+acclim}$ ’ simulations and all eight biomes, these are less than MODIS-based measurements. The two set of simulations have similar GPP estimates, illustrating weak indirect couplings in the JULES model between respiration changes and influence (e.g. via hydrological cycle) on gross primary productivity.”

New Figure 5 (left) has caption: “**Figure 5. Data and model-based global estimates of NPP and GPP for different biomes.** (a) Global measurements of total annual mean NPP, average for years 2000-2011, and using Earth-observed MODIS measurements. Values are spatially aggregated for different World Wildlife Fund (WWF) biome classifications, and their dominant type at each location is used in the “MOD17” algorithm applied to MODIS values (horizontal axis). Gridbox-mean JULES estimates of NPP are multiplied by gridbox area, and combined for each WWF biome (vertical axis). This is dependent upon which WWF biome is dominant for the grid location. Note logarithmic axes. JULES NPP estimates are slightly negative and so off axes for Mediterranean grasslands. (b) Similar calculation for GPP, with measurements from the “Model Tree Ensemble; MTE” algorithm. Both panels, model values presented in blue for standard JULES version (i.e. ‘standard’ simulation), and in red for new $R_{d,25}$ values with “b,c” temperature response and acclimation (i.e. ‘new $R_{d,25}+b,c+acclim$ ’ simulation). For GPP, differences are small between two model forms,

with red symbols overlapping blue symbols.”

In Methods we write: “In Figure 5, we present data and model-based estimates of global NPP and GPP, divided into eight biomes that are in turn based on 13 in the WWF definitions of terrestrial ecoregions³⁹. This reduction is by merging “tropical”, “subtropical forests” and “mangroves” into “tropical forests”; merging “temperate mixed-forests” and “temperate conifers” into “(extratropical) mixed forests”, and merging “temperate grasses”, “flooded grasses” and “montane grasses” into “temperate grassland”.

While the authors are right that more extensive dataset to estimate parameters IMPLIES an improvement in the model, it does not provide evidence for that the global estimates of respiration have been improved. The authors chose to limit any comparisons with data to a few locations:

“Figure 3 shows model output for single illustrative locations.....We select multiple South American sites, as these are of the few where measurements are available of all respiration components.”

GlobResp data merged in to our modelling structure allows us to get as far as we possibly can in understanding the role of respiration in the global carbon cycle. For upper-canopy levels, it is a model improvement. However we fully recognise there are still uncertainties, and particularly how upper canopy R_d understanding should be aggregated to other components. We hope our paper (and including new Figure 5) will provide a strong incentive for the research community to make these next steps in

building the necessary datasets. In part due to this reviewer request, the manuscript is now written very carefully so as to avoid risk of over-selling its findings.

New Figure 5 provides large-scale geographical information, and so complements the point analysis of Figure 3.

This could provide some support to their case but a statistical assessment of the modeled respiration against the measurements is not reported - nor are there any uncertainties presented for the data. (from Mahli et al the uncertainty for R_{leaf} is 2.84 Mg C ha⁻¹ year⁻¹ for Tambopata 05 and 2.07 Mg C ha⁻¹ year⁻¹ for Tambopata 06).

Uncertainty bounds have now been placed in Figure 3. Please see new diagram version below. We achieved this by returning to the *GlobResp* database and extracting the uncertainty bounds on parameters r_0 , r_1 and r_2 . We have then propagated this uncertainty through to all components of canopy respiration. This then allows formal confidence limits to be placed on site-based model projections. Furthermore, it enables direct comparison against the uncertainty bounds on measurements, where these are extracted from the literature.

After altering one component of the model, there is no reason not to show an evaluation of the model's estimates of total respiration or other fields against one or many of the established

benchmarks for LSMs. If the model has improved the leaf respiration then the model should perform adequately against total ecosystem respiration, if it doesn't that would indicate a problem. Leaving out this information just leaves the reading guessing.

Our new Figure 5, along with enhanced discussion and listing of caveats, should now remove any potential guessing by a reader. We feel we have improved significantly modelled upper-canopy respiration. However, when we propagate this through a typical in-canopy aggregation scheme, then NPP values are lower than expected. By now being explicit about this (e.g. via Figure 5), we hope this will encourage the community to establish (i) measurement campaigns similar to GlobResp, but for lower canopy levels and including stems and roots, and (ii) place any needed additional constraints on modelled GPP.

Despite this caution, however, in our Figure 3 then for the South American sites at least (i.e. where comprehensive data is available) then there is evidence that our new formulation performs better for canopy respiration $R_{d,c}$ and whole-plant respiration R_p . We appreciate the reviewer request for error bars in that figure, which helps to formalise that comparison.

I would have expected the paper to flow as follows 1) improved model structure 2) parameterization 3) rigorous validation at a limited number of sites and 4) broad comparison with established benchmarks 5) implications. As the paper stands step 3 is not rigorous, step 4 is absent and so the implications are not effectively demonstrated. The conclusions are therefore highly tempered in this version.

From the requests above, uncertainty bounds on Figure 3 brings more rigour to the site-specific measurements (point (3)), whilst new Figure 5 and related text introduces benchmarks based on Earth Observation (point(4)).

Reviewer #2 (Remarks to the Author):

Dear Reviewer 2. As for Reviewer 1, can we thank you for your time assessing our analysis, and especially as you have reviewed it more than once. Your requests have strong similarities to those of Reviewer 1. We show again the new diagrams below, just in case this letter is split in to individual components to each reviewer.

We have answered all comments. Please find our replies in blue, and indented.

The article “Implications of improved representations of plant respiration in a changing climate” by Huntingford et al. examines the carbon cycle impact of revising the temperature responses of plant respiration within the JULES land surface model. The authors find that the new representations of basal rate respiration increase global plant respiration compared to the standard JULES model, an effect that is partially offset by adding acclimation responses. This is the second Nature journal I have reviewed this article for. I still feel that this is an important and timely study. However, I have many of the same reservations, which I don't

feel were adequately addressed from the previous iteration of the manuscript I reviewed. Unfortunately, the authors were not able to present a response to previous reviews to explain how and why they addressed my previous concerns. My main concern is with the implications of the revised formulas. I feel they are overly broad and should be rephrased. I lay out my reasoning below. To summarize, I am concerned that the main results are very model specific, which is okay, but needs to be clearly stated. I am also not convinced that the revised formulations provide substantial model improvements at the large scales at which the conclusions are drawn. The sparse model-data comparison presented is great, but certainly does not convince me that the formulations are better.

We have addressed the three main points raised as follows. (1) “Concerned the main results are very model specific”. We have reviewed the literature of ten other mainstream land surface models. We find six of these have a direct respiration dependence on Nitrogen, whilst two others are implicit via a $V_{c,max}$ dependence. This suggests our findings of mapping GlobResp on to the JULES model will be representative of other land surface models. We now write in the main paper:

“Our analysis has implications for other ecosystem modelling groups. From a survey of ten leading land surface models, six of these simulate leaf respiration with a dependency on nitrogen content (models listed in Methods).”

In Methods we write: *“A review of the dependencies of ten other major land surface models shows that for six of these, upper canopy leaf respiration is dependent on leaf nitrogen content. The dependences are: BETHY is $V_{c,max}$ ⁵²; BIOME3 is $V_{c,max}$ ³⁴; BIOME-BGC is Nitrogen⁵⁴; Century is Nitrogen⁵⁵; CLM is Nitrogen⁵⁶; LPJ is Nitrogen²⁸; O-CN is Nitrogen⁵⁷; Orchidee is Empirical⁵⁸; Sheffield-DGVM is Nitrogen⁵⁹ and TEM is Empirical⁵⁵. The two models with dependence on $V_{c,max}$ contain an implicit dependence on nitrogen, via assumed $V_{c,max}$ -N relationships.”*

- 52 Ziehn, T., Kattge, J., Knorr, W. & Scholze, M. Improving the predictability of global CO₂ assimilation rates under climate change. *Geophysical Research Letters* **38**, doi:10.1029/2011GL047182 (2011).
- 53 Haxeltine, A. & Prentice, I. C. A general model for the light-use efficiency of primary production. *Functional Ecology* **10**, 551-561, doi:10.2307/2390165 (1996).
- 54 White, M. A., P.E., T., S.W., R. & R.R., N. Parameterization and Sensitivity Analysis of the BIOME-BGC Terrestrial Ecosystem Model: Net Primary Production Controls. *Earth Interactions* **4**, 1-85, doi:10.1175/1087-3562(2000)004<0003:pasaot>2.0.co;2 (2000).
- 55 Melillo, J. M. *et al.* Global climate-change and terrestrial net primary production. *Nature* **363**, 234-240, doi:10.1038/363234a0 (1993).
- 56 Lawrence, D. M. *et al.* Parameterization Improvements and Functional and Structural Advances in Version 4 of the Community Land Model. *J. Adv. Model. Earth Syst.* **3**, 27, doi:10.1029/2011ms000045 (2011).
- 57 Zaehle, S. & Friend, A. D. Carbon and nitrogen cycle dynamics in the O-CN land surface model: 1. Model description, site-scale evaluation, and sensitivity to parameter estimates. *Global Biogeochemical Cycles* **24**, 13, doi:10.1029/2009gb003521 (2010).
- 58 Krinner, G. *et al.* A dynamic global vegetation model for studies of the coupled atmosphere-biosphere system. *Global Biogeochemical Cycles* **19**, 44, doi:10.1029/2003gb002199 (2005).

Comment (2) “I am also not convinced that the revised formulations provide substantial model improvements at the large scales at which the conclusions are drawn”. The *GlobResp* dataset placed in a gridded modelling framework provides a step change in predictions of upper canopy respiration. However, at present and in the absence of parallel datasets for other ecosystem respiration components, then the whole-canopy projections can only be regarded as illustrative. To make this point clearer, we have developed a new Figure 5. This compares over large scales for different biome regions, our JULES estimates of NPP (and GPP) against satellite retrievals. It highlights differences that remain open research questions, and that we hope the community might move forward to answer.

We write: “.. in Figure 5 we perform large-scale comparisons against two Earth Observation-based datasets. These are estimates of NPP from the MODIS satellite, using the MOD17 algorithm^{36, 37}, and of GPP from the Model Tree Ensemble (MTE) method³⁸. Both of these algorithms evaluate mean NPP and GPP values depending on location, and are mapped on to local dominant biomes in turn based on the World Wildlife Fund (WWF) ecoregion classifications³⁹ (Methods). These data-based estimates, locally, represent mean NPP and GPP, and so for parity we compare against modelled gridbox mean JULES calculations of the equivalent fluxes. That is, we use areal weighting of the five PFT types in JULES for each position. To keep equivalence with the WWF categories, we plot in Figure 5 total annual NPP and GPP for both data and JULES, integrated over areas for the biomes as marked. Presented are ‘standard’ and ‘new $R_{a,25+b,c+acclim}$ ’ simulations. Calculations with ‘new $R_{a,25}$ ’ and ‘new $R_{a,25+b,c}$ ’ model format are very similar to ‘new $R_{a,25+b,c+acclim}$ ’ and so not shown. As expected, in all cases, introduction of GlobResp-based respiration estimates results in significantly lower modelled NPP values. Furthermore for ‘new $R_{a,25+b,c+acclim}$ ’ simulations and all eight biomes, these are less than MODIS-based measurements. The two set of simulations have similar GPP estimates, illustrating weak indirect couplings in the JULES model between respiration changes and influence (e.g. via hydrological cycle) on gross primary productivity.”

Our new Figure 5 (left) has caption: “**Figure 5. Data and model-based global estimates of NPP and GPP for different biomes.** (a) Global measurements of total annual mean NPP, average for years 2000-2011, and using Earth-observed MODIS measurements. Values are spatially aggregated for different World Wildlife Fund (WWF) biome classifications, and their dominant type at each location is used in the “MOD17” algorithm applied to MODIS values (horizontal axis). Gridbox-mean JULES estimates of NPP are multiplied by gridbox area, and combined for each WWF biome (vertical axis). This is dependent upon which WWF biome is dominant for the grid location. Note logarithmic axes. JULES NPP estimates are slightly negative and so off axes for Mediterranean grasslands. (b) Similar calculation for GPP, with measurements from the “Model Tree Ensemble; MTE” algorithm. Both panels, model values presented in blue for standard JULES version (i.e. ‘standard’ simulation), and in red for new $R_{d,25}$ values with “b,c” temperature response and acclimation (i.e. ‘new $R_{d,25}+b,c+acclim$ ’ simulation). For GPP, differences are small between two model forms,

with red symbols overlapping blue symbols.”

In Methods we write: “In Figure 5, we present data and model-based estimates of global NPP and GPP, divided into eight biomes that are in turn based on 13 in the WWF definitions of terrestrial ecoregions³⁹. This reduction is by merging “tropical”, “subtropical forests” and “mangroves” into “tropical forests”; merging “temperate mixed-forests” and “temperate conifers” into “(extratropical) mixed forests”, and merging “temperate grasses”, “flooded grasses” and “montane grasses” into “temperate grassland”.

Comment (3): “The sparse model-data comparison presented is great, but certainly does not convince me that the formulations are better”. To help towards this, uncertainty bounds in revised Figure 3 (presented below) are now placed on both the site-specific model projections, and for the measurements. This allows better comparison. We achieve this by returning to the *GlobResp* database and extracting the uncertainty bounds on parameters r_0 , r_1 and r_2 . We then propagate this uncertainty through to all components of canopy respiration. This then allows formal confidence limits to be placed on projections for different respiration components. Uncertainty bounds on measurements are extracted from the literature.

In the Figure 3 below, for the South American sites, there is evidence that the new formulation performs better for canopy respiration $R_{d,c}$ and whole-plant respiration R_p .

- The main conclusion from the abstract is that revised respiration estimates could be 30% higher than previous estimates. The previous estimates were from a single model (JULES). While this is interesting, it is highly model specific and the effect is likely to differ, possibly in both magnitude and direction if compared to other models. This caveat needs to be explicitly stated throughout, including in the abstract. In my opinion, the more interesting comparisons are to the different revised formulations, which have greater commonalities, rather than the comparison to the standard JULES.

For our answer to this, please see our response above starting: “We have addressed the three main points raised as follows”, and in particular the use of literature to determine the form of R_d components in the ten other leading land surface models. In our view, given the strong similarities, then we would expect *GlobResp* data to create very similar patterns of change in those land modelling systems. Please also see our response two comments below this.

- Following from the previous comment, the main result stems primarily from a change in the basal rate respiration calculation within JULES, which involved a couple of important steps: (1) the switch from a R_d - V_{cmax} to R_d - N relationship and (2) the implementation of GLOBRESP parameterizations. Do the authors have a sense for which of these is driving the effect? The sense I get from reading the manuscript is that the latter effect is dominant,

although I would think that former may be quite important, primarily due to the uncertainty in estimating leaf/whole plant nitrogen contents throughout the canopy.

We answer this directly by an analysis of scale. In the main paper we write: “The relative importance of contributions (Methods) to revised $R_{d,25}$ values are broad changes to overall baseline having most influence (via parameters r_0 , r_1 and r_2 considered together), followed by the specific acclimation dependency and then the relationship with leaf nitrogen.”

Then in the Methods, the analysis gives new sentences: “*The relative importance of contributions to revised $R_{d,25}$ can be assessed from Tables M1 and M2. In general terms, and for broadleaf trees, the new representative $R_{d,25}$ values change from 0.4157 to 1.136 $\mu\text{mol CO}_2\text{m}^{-2}\text{s}^{-1}$. From the TRY database¹⁸, then with 80% confidence leaf nitrogen concentrations lie between 62% and 154% of their median value. This gives a range of $0.237 < r_{1n_{l,a}} < 0.593 \mu\text{mol CO}_2\text{m}^{-2}\text{s}^{-1}$. Growth temperature ranges of 5 °C to 25 °C give $0.2 < r_2T_G < 1.0 \mu\text{mol CO}_2\text{m}^{-2}\text{s}^{-1}$. This simple scale argument suggests a decreasing importance, both in terms of absolute and potential variability, of contributions to new $R_{d,25}$ as new baseline, followed by acclimation and then leaf nitrogen dependence.*”

- To address the model specificity problem, the authors could add lines from other land surface models to Figure 1.

We did consider this possibility. It has taken us significant time to implement *GlobResp* within the JULES model, and to then link that structure to outputs from 34 GCMs (to capture climate uncertainty). It would be a particularly large task to link with the ten other main land surface modelling groups, and implement these changes for each of them. We respectfully ask that for now, we recognise from the literature our findings are likely to have strong similarities should *GlobResp* eventually be used by other land modelling groups. However, based on this comment, we have written a strong “Data and Code Availability” paragraph. We will make all information leading to our five Figures fully available. This, in parallel with the paper itself, could encourage the launching of a “Respiration MIP” at some future date.

- From the model-data comparisons, I am not convinced that the revised formulas work better for simulating plant respiration at large scales, which undermines the conclusions a bit. I am sympathetic with the fact that respiration at large scales are sparse, but even in the few sites examined the results are quite mixed, with some estimates being improved at some sites, but not others and the improvement within sites dependent on the estimate evaluated. I think these comparisons should be given a greater emphasis in the study, particularly over the broad and uncertain conclusions from the global simulations.

In light of this request (and similar from Reviewer 1), we wanted to achieve more in terms of checking performance at the global scales. This has led to the new Figure 5, and associated text and discussion. Please see above where this new diagram has been copied in to this text.

Additionally, as the few available sites where direct comparison is allowed, we have now added uncertainty bounds both on model projections and – from the literature – on the measurements too. The revised Figure 3 with these bounds is also copied above in to this text.

- I would like to see a greater discussion of the empirical nature of plant respiration formulations in land surface models such as JULES. While the dataset used for the new parameterizations is large relative to its predecessors, it is quite limited in space and time compared to the data simulated by the global simulations. This is a problem given that the mechanisms underlying the respiration responses to temperature and leaf N are not explicitly simulated, but rather assumed from empirical relationships, which is an issue when extrapolating to larger scales. This study is a clear step forward for improving the empirical relationships, but it should be made clearer that these still suffer from a lack of a mechanistic incorporation of plant respiratory responses to temperature and nitrogen.

Based on this request, we have enhanced a couple of previous sentences that briefly mentioned the eventual need for mechanistic models. These are now expanded, and to give them prominence, are now placed in the Discussion. We write: “*Full mechanistic models, which can still be tested against GlobResp data, ultimately may allow further advances on empirical-based descriptions of respiration. However, availability of these remains a long way from routine usage, yet alone in large-scale climate models. This is an issue recently discussed in depth for the “b,c” instantaneous temperature response formulation^{40, 41}, and where that exchange in the literature has relevance to more general respiration modelling.*”

40 Adams, M. A., Rennenberg, H & Kruse, J. Different models provide equivalent predictive power for cross-biome response of leaf respiration to temperature. *Proceedings of the National Academy of Sciences of the United States of America* **113**, E5993-E5995, doi:10.1073/pnas.1608562113 (2016).

41 Heskell, *et al.* Reply to Adams et al.: Empirical versus process-based approaches to modeling temperature responses of leaf respiration. *Proceedings of the National Academy of Sciences of the United States of America* **113**, E5996-5997, doi:10.1073/pnas.1612904113 (2016).

Reviewers' Comments:

Reviewer #1:

Remarks to the Author:

What are the major claims of the paper?

This paper claims there is substantial uncertainty in current estimates of plant respiration and that this issue deserves the urgent attention of the global land surface modeling community and the physiological ecologist and ecosystem community because revised estimates using the JULES model indicate a 30% change in plant respiration. Should this magnitude of change be common across other models, it's likely that the land carbon sink estimated by LSMs is overestimated.

The work demonstrates that for this model, changes to the plant respiration routines result in plausible changes in NPP and GPP at the ecosystem scale and also across biomes. This change is consistent with an increase in plant respiration (i.e. the NPP is lower than the standard model). With this global analysis - the authors have satisfied my earlier concerns and have provided other modeling groups with a benchmark to compare their own work. I hope that the authors will consider making their global runs freely and openly available to allow other groups to compare. I'm really excited to see this comparison.

The magnitude of a 30% change in plant respiration is likely to be particular to the JULES model - however in other models this number could be larger or smaller. Because other models share similar logic in their representations of Plant respiration, it is prudent for other groups to test out changes similar to those proposed here. This work may well prompt other Land Surface Modeling groups to implement similar plant respiration routines based on the very useful Globresp database.

Reviewer #2:

Remarks to the Author:

In their revised article titled "Implications of improved representations of plant respiration in a changing climate" by Huntingford et al. have addressed the concerns raised by myself and another reviewer. I appreciate the revision of Figure 3 and the addition of Figure 5.

With the new Figure 3, the authors have supported their claim that new formulations provide better respiration estimates, though I am concerned that no statistical analysis was done. Also, it seems clear that new formulations worsen NPP estimates (apparent in Figure 5 as well), suggested to be due to GPP in Figure 3.

When looking at Figure 5, however, it seems that GPP estimates are okay (near 1:1), suggesting that the increased error in NPP estimates (Figure 5a) is actually due to poor estimation of respiration, which contradict the findings in Figure 3! Is there an explanation for this apparent difference?

In my opinion, it is not entirely necessary that the new model perform "better," as the data used for parameterization are clearly more comprehensive and the data available for benchmarking are far from perfect. However, the discussion of reductions in future CO₂ mitigation and increased possibility of ecosystem die-back are not warranted from this study and I would strongly caution the authors from making such claims without far more rigorous (and convincing!) benchmarking.

Small notes:

- Why are there no error bars on Figure 5?
- Wouldn't Figure 5 present better using spatial maps?

We have addressed all reviewer requests, and as described below. Please find our replies in blue and indented.

Reviewers' comments:

Reviewer #1 (Remarks to the Author):

Dear Reviewer 1. We thank you once again for your time in helping us with this manuscript. Your comments and suggestions have continued to aid the manuscript.

Reviewer #1 (Remarks to the Author):

What are the major claims of the paper?

This paper claims there is substantial uncertainty in current estimates of plant respiration and that this issue deserves the urgent attention of the global land surface modeling community and the physiological ecologist and ecosystem community because revised estimates using the JULES model indicate a 30% change in plant respiration. Should this magnitude of change be common across other models, it's likely that the land carbon sink estimated by LSMs is overestimated.

The work demonstrates that for this model, changes to the plant respiration routines result in plausible changes in NPP and GPP at the ecosystem scale and also across biomes. This change is consistent with an increase in plant respiration (i.e. the NPP is lower than the standard model). With this global analysis - the authors have satisfied my earlier concerns and have provided other modeling groups with a benchmark to compare their own work. I hope that the authors will consider making their global runs freely and openly available to allow other groups to compare. I'm really excited to see this comparison.

I can confirm that we have made all simulations available. These are on the Environmental Information Data Centre (EIDC), which is a recognised repository by Nature series journals. The model output files are described in full, and via an assigned doi. This doi is now given in the paper.

The magnitude of a 30% change in plant respiration is likely to be particular to the JULES model - however in other models this number could be larger or smaller. Because other models share similar logic in their representations of Plant respiration, it is prudent for other groups to test out changes similar to those proposed here. This work may well prompt other Land Surface Modeling groups to implement similar plant respiration routines based on the very useful Globresp database.

Please see comment above. We are happy for all our simulations to be available for others to compare against alternative land simulations, and as respiration components are progressed in them. All data is available at the following doi:
<https://doi.org/10.5285/24489399-5c99-4050-93ee-58ac4b09341a>

In the section Data and Code Availability, we now add: “All JULES model outputs, and for the four factorial experiments, are available for download from the Environmental Information Data Centre. The address is:
<https://doi.org/10.5285/24489399-5c99-4050-93ee-58ac4b09341a>”

Reviewer #2 (Remarks to the Author):

Dear Reviewer 2. We thank you once again for your time in helping us with this manuscript. Your comments and suggestions have continued to aid the manuscript.

In their revised article titled “Implications of improved representations of plant respiration in a changing climate” by Huntingford et al. have addressed the concerns raised by myself and another reviewer. I appreciate the revision of Figure 3 and the addition of Figure 5.

Thank you.

With the new Figure 3, the authors have supported their claim that new formulations provide better respiration estimates, though I am concerned that no statistical analysis was done. Also, it seems clear that new formulations worsen NPP estimates (apparent in Figure 5 as well), suggested to be due to GPP in Figure 3.

Figure 3 has uncertainty bounds placed on modelled respiration components and on site measurements of respiration, GPP and NPP. The model-based uncertainty bounds are related to those from Atkin et al (2015), whilst the site-based estimates are from the papers associated with each measurement campaign. The bounds are built differently, precluding more standard t -tests to compare model versus data means.

However, we accept the earlier manuscript version has not utilised the statistical bounds well. To assess model ability, we adopt a simple measure that a model performs well if its projected value falls within the uncertainty bounds of the data.

To address this, we now make clear statements about point sites, writing in the paper: “*More specifically, we define the JULES model as having improved performance when the standard simulation estimate of R_p lies outside the data-based bounds on whole-plant respiration, but simulations new $R_{d,25+b,c+acclim}$ then fall within those bounds. This happens for the sites at Manaus, Tambopata, Iquitos (dataset ‘a’), and Guarayos (dataset ‘a’).*”

When looking at Figure 5, however, it seems that GPP estimates are okay (near 1:1), suggesting that the increased error in NPP estimates (Figure 5a) is actually due to poor estimation of respiration, which contradict the findings in Figure 3! Is there an explanation for this apparent difference?

In response to reviewer comment below, we now present the global mean NPP and GPP estimates also as maps (Figure 6). This is for both data and model estimates. This illustrates that although GPP model estimates for Tropical Forests (TF) are relatively near to the 1:1 line at the global scale (Figure 5), there are strong regional differences. This includes for the Amazon basin, where the model underestimates GPP – yet there are other regions where GPP is overestimated. This explains better the South American point findings of Figure 3.

We now write as explanation in the text “*In Figure 6, we add geographical information to our global data estimates of NPP and GPP, and for corresponding*

JULES simulations with all effects i.e. “new $R_{d,25+b,c+acclim}$ ” (expanding on the red symbols of Figure 5). Figure 6a is “MOD17-based” NPP estimates, and Figure 6b is JULES NPP estimates. In general, modelled NPP is smaller across all geographical points. For GPP, the situation is slightly less clear. In most regions (Figure 6c versus Figure 6d), then “MTE-based” GPP estimates are again higher than those of the JULES model. This includes much of the Amazon region. However, for the tropics, some modelled GPP values are actually higher than data. This offers an explanation as to why GPP appears underestimated in some tropical points of Figure 3, yet for the average across Tropical Forest (TF), JULES performs well (Figure 5b).”

In my opinion, it is not entirely necessary that the new model perform “better,” as the data used for parameterization are clearly more comprehensive and the data available for benchmarking are far from perfect. However, the discussion of reductions in future CO₂ mitigation and increased possibility of ecosystem die-back are not warranted from this study and I would strongly caution the authors from making such claims without far more rigorous (and convincing!) benchmarking.

We fully accept this.

We are still keen to make reference to the fact that respiration changes under global warming could affect emissions reductions needed to avoid crossing key temperature targets (e.g. two degrees), and severe respiration losses could ultimately trigger biome changes. We believe our comprehensive *GlobResp* database helps constrain top-leaf level respiration, but there is still research to be performed (alongside extensive benchmarking) before a definitive answer can be given as to whole-canopy respiration interactions.

We have therefore taken four actions on this:

- (1) We have removed the last words from the Abstract, which read “potentially reducing future natural mitigation of CO₂ emissions”.
- (2) Sentence in Discussion on mitigation re-worded less strongly to now simply read: “As global land-atmosphere CO₂ fluxes are a small difference between large fluxes, future terrestrial ecosystem respiration responses to warming can therefore influence the natural ability to offset CO₂ emissions.”
- (3) Sentence in Discussion on biome change now just reads: “If future increases in respiration overtake any thermal or CO₂-ecosystem fertilisation, lower NPP values in the most extreme instances could force biome changes”
- (4) Re-iterated the need for more extensive with-canopy data for benchmarking. Following Discussion sentence starting “Equivalent global respiration measurement campaigns to *GlobResp*, but for other canopy components” we now add: “Such additional data will enable more rigorous benchmarking of different terrestrial model configurations of within-canopy respiration fluxes”

Small notes:

- Why are there no error bars on Figure 5?

Although the GPP data does have regional error estimates, there are no local error estimates for NPP. However for both quantities, the peer-review literature does give

more general large-scale global estimates of $\pm 15\%$ for NPP (Ito 2011) and $\pm 7\%$ for GPP (Beer et al. 2010: Global GPP=123 \pm 8 Pg C yr⁻¹). For consistency, we select these ranges for our global biomes. These values are now in Figure 5, shown as horizontal error bars, and for the $R_{d,25}+b,c+acclim$ case (red symbols).

We now write in the manuscript: “Uncertainly bounds on data take identically the global literature values of $\pm 15\%$ for NPP³⁷ and $\pm 7\%$ for GPP³⁸. These are horizontal black bars, shown only on *new* $R_{d,25}+b,c+acclim$ points.”

[37] Ito, A. A historical meta-analysis of global terrestrial net primary productivity: are estimates converging? *Global Change Biology* **17**, 3161-3175, doi:10.1111/j.1365-2486.2011.02450.x (2011).

[38] Beer, C. et al., 2010. Terrestrial Gross Carbon Dioxide Uptake: Global Distribution and Covariation with Climate. *Science*, 329(5993): 834-838.

- Wouldn't Figure 5 present better using spatial maps?

We are grateful for this request, as it has aided understanding of how our model performs geographically as well as globally. We have decided to present this as an additional Figure 6. Below we present the new figure, and its caption.

Figure 6 description in the revised paper itself is given above, in response to comment starting “When looking at Figure 5...”. The model estimates in Figure 6 are with all changes included – i.e. new $R_{d,25}$, improved instantaneous temperature response and acclimation.

Figure 6. Data- and model-based maps of estimates of NPP and GPP for different biomes. (a) Map of annual mean NPP, average for year 2000-2011 and using the “MOD17” applied to MODIS values. Land points included have less than 50% agriculture. (b) JULES estimates for the same gridboxes and period as (a), and with model configuration ‘*new $R_{d,25+b,c+acclim}$* ’. (c) Map of annual mean GPP, same period and land points as (a). This is based on measurements from the “Model Tree Ensemble; MTE” algorithm. (d) JULES estimates of GPP, again with ‘*new $R_{d,25+b,c+acclim}$* ’ form, and same time and land points as (b). Panel (e) shows the ecoregion classifications, and so the areally averaged values across these for panels (a)-(d) are the red symbols in Figure 5. Panel (e) labels identical to Figure 5.

Reviewers' Comments:

Reviewer #2:

Remarks to the Author:

In their revised article, the authors have addressed the concerns raised by myself and another reviewer. As a result, the manuscript is much improved and I would recommend publication.

Minor comments

Figure 3g: Data are plotted outside of the range of the graph.

Figure 6: Combining panels a and b as well as panels c and d to show the percent difference between the data and model would make the figure easier to interpret.

Dear Reviewer

Thank you again for your help regarding our manuscript: **“Implications of Improved Representations of Plant Respiration in a Changing Climate”**. We do appreciate your substantial time now spent on this paper.

The problem with the clipped error bar in Figure 3g has been corrected for. For Figure 6, we now present percentages, as requested. Below is the new diagram, the revised caption, and the associated text within the paper itself.

With kind regards,

Chris Huntingford

Figure 6. Data- and model-based maps of comparison of net primary productivity (NPP) and gross primary productivity (GPP) for different biomes. (a) Map of Joint UK Land Environmental Simulator (JULES) estimates of annual NPP, average for year 2000-2011 divided by MODerate-resolution Imaging Spectroradiometer (MODIS) NPP algorithm (MOD17) estimates for the same period. Values multiplied by one hundred to express as percentage. Land points excluded are those with more than 50% agriculture, and also where values are very small (if absolute value of

JULES or MODIS NPP is less than $1 \text{ gC m}^{-2} \text{ yr}^{-1}$). **(b)** Similar to (a) but for GPP, and data based on upscaled FLUXNET GPP from the Model Tree Ensemble (MTE) algorithm. Again, land points excluded are those with more than 50% agriculture, and those with small values (if value of JULES or MODIS GPP is less than $1 \text{ gC m}^{-2} \text{ yr}^{-1}$). Panel **(c)** is map of dominant biome, and labels identical to Figure 5.

In Figure 6, we add geographical information to our global data estimates of NPP and GPP, and for corresponding JULES simulations with all effects, i.e. $\text{New_}R_{d,25_b,c_acclim}$ (expanding on the red symbols of Figure 5). Figure 6a is JULES NPP estimates divided by MOD17-based NPP estimates (and multiplied by 100 to give percentage). In general modelled NPP with new plant respiration description, is smaller than MOD17 NPP across the geographical points. For some points it can give unsustainable negative modelled NPP values. For GPP, the situation is slightly less clear. Figure 6b is JULES GPP estimates divided by MTE-based GPP values, again as percentage. For many points, the JULES model is also underestimating GPP, and this includes much of the Amazon region. However, for the tropics, a few modelled GPP values are actually higher than data. This offers an explanation as to why GPP appears underestimated in some tropical points of Figure 3, yet for the average across Tropical Forest (TF), JULES performs well (Figure 5b). Figure 6b also shows that modelled GPP is usually too low outside of the tropics. This is why, when combined with the enhanced respiration of $\text{New_}R_{d,25_b,c_acclim}$ formulation, this can lead to very low or even unsustainable negative NPP. Figure 6c shows the dominant WWF-defined biomes for each location.

[Figure 6 here]